# Unsupervised Discovery of Formulas for Mathematical Constants

**Michael Shalyt**[*]  **Uri Seligmann**[*]  **Itay Beit Halachmi**   **Ofir David**
**Rotem Elimelech**   **Ido Kaminer**
Technion - Israel Institute of Technology, Haifa 3200003, Israel
shalyt@technion.ac.il, uri.seligmann@gmail.com
itaybe@campus.technion.ac.il, eofirdavid@gmail.com
rotem.eli@campus.technion.ac.il, kaminer@technion.ac.il

## 1   Abstract

Ongoing efforts that span over decades show a rise of AI methods for accelerating scientific discovery [Fajtlowicz, 1988, Petkovsek et al., 1996, Wolfram et al., 2002, Buchberger et al., 2006, Bailey et al., 2007, Raayoni et al., 2021, Davies et al., 2021, Fawzi et al., 2022], yet accelerating discovery in mathematics remains a persistent challenge for AI. Specifically, AI methods were not effective in creation of formulas for mathematical constants because each such formula must be correct for infinite digits of precision, with "near-true" formulas providing no insight toward the correct ones. Consequently, formula discovery lacks a clear distance metric needed to guide automated discovery in this realm. In this work, we propose a systematic methodology for categorization, characterization, and pattern identification of such formulas. The key to our methodology is introducing metrics based on the convergence dynamics of the formulas, rather than on the numerical value of the formula. These metrics enable the first automated clustering of mathematical formulas. We demonstrate this methodology on Polynomial Continued Fraction formulas, which are ubiquitous in their intrinsic connections to mathematical constants [Lagarias, 2013, Bowman and McLaughlin, 2002, Laughlin and Wyshinski, 2004], and generalize many mathematical functions and structures. We test our methodology on a set of 1,768,900 such formulas, identifying many known formulas for mathematical constants, and discover previously unknown formulas for $\pi$, $\ln(2)$, Gauss', and Lemniscate's constants. The uncovered patterns enable a direct generalization of individual formulas to infinite families, unveiling rich mathematical structures. This success paves the way towards a generative model that creates formulas fulfilling specified mathematical properties, accelerating the rate of discovery of useful formulas.

## 2   Introduction

Historically, formulas of mathematical constants were a symbol of aesthetics and beauty. Continued fraction formulas such as those for the golden ratio $\phi$ and for $\tan(x)$

$$1 + \cfrac{1}{1 + \cfrac{1}{1 + \cfrac{1}{1 + \cdots}}} = \phi \quad \cfrac{x}{1 - \cfrac{x^2}{3 - \cfrac{x^2}{5 - \cdots}}} = \tan(x) \tag{1}$$

enable calculating infinitely many digits for these constants. Discovering such formulas often leads to profound revelations regarding the properties and underlying structure of fundamental constants. For example, the continued fraction formula for $\tan(x)$, shown in Eq. 1, was used by Johann Heinrich Lambert in the first proof of the irrationality of Pi [Lambert, 1768][Berggren et al., 2004]. Unfortunately, such formulas are notoriously hard to find on-demand, often relying on a mathematician's profound intuition. Part of the challenge is the lack of a well-defined 'distance', or a

---

[*]Equal contribution.
Project repository: https://github.com/RamanujanMachine/Blind-Delta-Algorithm

38th Conference on Neural Information Processing Systems (NeurIPS 2024).

metric, between a formula and a given constant. i.e., there is no known way to tell whether a formula is nearly accurate. The formula either works, or it does not. In other fields of science, a prediction accurate to 1000 digits is precise enough for any practical need. However, in mathematics, if the 1001ˢᵗ digit is wrong, the formula is incorrect and gives no insight regarding a correct formula. This lack of a metric is a substantial hurdle both for human efforts and for automated analysis, as many methods for optimization such as gradient descent become unsuitable.

Recent efforts developed computer algorithms to discover a multitude of formula hypotheses for mathematical constants [Raayoni et al., 2021], even implementing the first large-scale distributed computation for such discoveries [Elimelech et al., 2023], but they relied mostly on exhaustive search methods. These approaches complements earlier applications of algorithms for automated theorem proving (ATP) (such as computer proofs of hypergeometric identities [Petkovsek et al., 1996], Malarea [Urban, 2007], and Flyspeck [Kaliszyk and Urban, 2012]), and automated conjecture generation (ACG) (such as mechanical mathematics [Wang, 1960], the Automated Mathematician [Lenat, 1982], EURISKO [Lenat and Brown, 1984, Davis and Lenat, 1982], and Graffiti [Fajtlowicz, 1988]).

Here we propose a fundamentally new methodology for automated investigation and discovery of formulas for mathematical constants. We constructed a large dataset of continued fractions, and enriched it with metrics based on their convergence dynamics, which are found to embody fundamental information about each continued fraction. These dynamical metrics enable the identification and generalization of patterns within the dataset. Using the metrics, we develop a process of categorization and clustering (Fig. 1) of continued fractions that share similar values of their dynamical metrics. Analyzing each automatically identified cluster of formulas, we find that all its members often relate to the same mathematical constant, showing the value of the dynamical metrics for the discovery of new formulas and the internal structure of families of such formulas. This novel method of formula discovery allowed us to identify both previously known and completely new formulas for constants such as $\pi$, $\ln(2)$, $\cot(1)$, the golden ratio, square roots of multiple integers, the Gauss constant, and the Lemniscate constant.

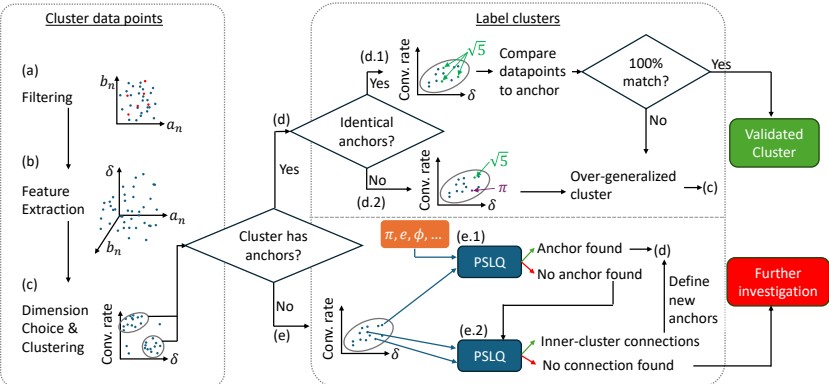

Figure 1: **Systematic clustering and labeling of formulas by dynamical metrics.** Our methodology analyzes Polynomial Continued Fractions (PCFs) in two main stages. **Clustering**: (a) Filter degenerate PCFs. (b) Evaluate PCFs and extract their dynamics-based metrics (section 3). (c) Choose the best few metrics and use them to cluster the data. **Labeling**: In every cluster, look for PCFs known in the literature and use them as anchors. (d) If anchors are found in the cluster, validate that they do not contradict, i.e., relate to different constants. (d.1) If all anchors are in agreement, choose a random subset of other points in the cluster and use PSLQ to validate that they also relate to the same constant. If the validation is successful, the cluster is labeled. If not, the cluster should be split. (d.2) If the anchors relate to different constants, the cluster should be split – return to step c for finer clustering of the data. When focusing on a specific cluster, the best metrics could be different than those for the full dataset. (e) If no anchor is found in a certain cluster, attempt to label by (e.1) choosing a small subset of PCFs in the cluster and running a PSLQ search for each of them against a large set of potential constants. If a connection is found, the cluster now has an anchor – return to step d. (e.2) If an anchor is still not found, attempt to connect a sample of data points within the cluster using PSLQ. If successful, conclude that the cluster is correct, but has no identified constant. Define a new label for that cluster. If PSLQ failed to connect points within the cluster, return to step c for finer clustering. If no further refinement is appropriate, flag the cluster for further analytical investigation.

As part of our analysis of metrics of continued fractions, we developed and applied the most complete classification of polynomial continued fractions known to date, detailed in Appendix B. This classification includes the prediction of whether the continued fraction converges directly based on its defining polynomials.

Traditional clustering methods attempt to relate data points by calculating distance metrics based on the parameters of these data points, e.g., the coefficients of the defining polynomials. The most common approaches (like SVM) rely on linear classification, while more advanced methods rely on non-linear kernel transformations - but usually use various functions calculated directly on the data parameters. In our dataset, each point is a continued fraction formula defined by the polynomials used to construct it. However, we find that it is not sufficient to use the parameters of the polynomials, and not even the numerical limit of the continued fraction. Instead, we find that it is the *dynamics* of the continued fraction generated by these polynomials, rather than any direct function on their coefficients, which provides the most useful metrics for analysis. In other words, we find that the useful underlying metrics to extract from each data point are embedded within the intricate progression of the sequence created by the formula, rather than the explicit numerical value (limit) of that formula, or the coefficients defining it. Thus, in order to assess the distance between two polynomial continued fractions, and identify relations between such formulas, it is imperative to characterize the nuanced behavior of their sequences, analyzing trends in the convergence process of these sequences, spanning over numerous terms.

Some of the metrics we extract, such as the irrationality measure, are well-known in the mathematical community, yet were never considered for a large-scale classification effort. The evaluation of the irrationality measure is technically challenging for formulas whose limit is not known in advance (which is the vast majority). This challenge made it impossible to extract the irrationality of formulas for a large dataset. Consequently, we develop a new algorithm – the Blind-$\delta$ algorithm (Section 3.4) – to enable the extraction of the irrationality measure of a continued fraction without prior knowledge of its limit. This algorithm allowed us to extract the irrationality measure for the entire dataset.

These advances provide the building blocks for our novel methodology for formula discovery. We cluster formulas by their 'closeness' to other formulas according to these new metrics, thus identifying promising formulas regardless of their numerical value (Fig. 1left). Once a candidate formula is found, we numerically validate it by calculating its value to a large precision and then identifying its relation to a mathematical constant. The "generate ⇒ validate" approach is inspired by works in AI-driven code generation [Ridnik et al., 2024] and problem solving in geometry [Trinh et al., 2024].

# 3 Methodology for Data-Driven Discovery

## 3.1 Definitions

*Polynomial Continued Fractions*

In this work we chose to focus on polynomial continued fraction (PCF) formulas as our test case due to the combination of their simplicity and expressive power. PCFs relate to a wide range of mathematical fields, represent a variety of constants, are equivalent to infinite sums [Euler, 1748], and cover mathematical functions such as Bessel functions, trigonometric functions, integral families, widely used Taylor series, and generalized hypergeometric functions [Cuyt et al., 2008]. Thus, studying PCFs can provide insight into a plethora of mathematical objects and applications.

A PCF at depth $n$ is defined as:

$$a_0 + \cfrac{b_1}{a_1 + \cfrac{b_2}{\ddots + \cfrac{b_n}{a_n}}} = \frac{p_n}{q_n}, \tag{2}$$

where $a_n = a(n)$ and $b_n = b(n)$ are evaluations of polynomials with integer coefficients. The PCF value is the limit $L = \lim\limits_{n\to\infty} \frac{p_n}{q_n}$ (when it exists). The converging sequence of rational numbers $\frac{p_n}{q_n}$ provides an approximation of $L$, which is known as a Diophantine approximation.

*The Irrationality Measure of a Number*

While irrational numbers cannot be expressed using a simple quotient of integers, they can be approximated by them. Moreover, some approximations are "better" than others, and one way to evaluate their quality is by a quantity called the irrationality measure [Hardy et al., 1979].

We define the *irrationality measure of a sequence* $\frac{p_n}{q_n} \to L$ as the limit $\delta_n \to \delta$, with

$$\delta_n = \frac{-\log \left| L - \dfrac{p_n}{q_n} \right|}{\log |\tilde{q}_n|} - 1 \quad , \quad \tilde{q}_n = \frac{q_n}{\gcd(p_n, q_n)} \tag{3}$$

For every $L \in \mathbb{R}$, the *irrationality measure of $L$* is defined as the supremum of all possible $\delta$ for which there is a sequence of distinct rational numbers $\frac{p_n}{q_n} \to L$; $\frac{p_n}{q_n} \neq L$ that satisfies

$$\left| L - \frac{p_n}{q_n} \right| < \frac{1}{q_n^{1+\delta}}. \tag{4}$$

It is known that for irrational numbers this measure is $\geq 1$ (Dirichlet theorem for Diophantine approximations), and for rationals it is 0.

Note that the irrationality measure of $L$ is greater or equal to the irrationality measure of any specific sequence converging to the same $L$. While the irrationality measure of a sequence can be any number $\geq$ -1, the irrationality measure of its limit $L$ is always either 0 or $\geq 1$ [Church, 2019].

### 3.2 $\delta$-Predictor Formula

The classification of a large number of continued fraction formulas requires an efficient and accurate calculation of the irrationality measure $\delta$ for each formula. This calculation is challenging because it depends on the asymptotic behavior of the converging sequence, and because $\delta$ appears as an exponent of a large basis number. The $\delta$-Predictor formula that we present here provides a way around this challenge - requiring no specific knowledge about the convergence rate and trajectory, or even about the sequence limit itself:

$$\delta_{\text{predicted}} = \lim_{n \to \infty} \frac{n \cdot \log \left| \dfrac{\lambda_1(n)}{\lambda_2(n)} \right|}{\log |\tilde{q}_n|} - 1 \tag{5}$$

where $\lambda_1(n)$ and $\lambda_2(n)$ are the eigenvalues of the matrix $\begin{pmatrix} 0 & b_n \\ 1 & a_n \end{pmatrix}$, $|\lambda_1(n)| > |\lambda_2(n)|$.

This formula extends a hypothesis made in a previous work [David et al., 2021], which was limited to PCFs with $\deg(B) = 2\deg(A)$ and with a $\tilde{q}_n$ that grows exponentially. As we found in this work, Eq.5 works for any converging PCF. It was validated numerically and proven for the $\deg(B) = 2\deg(A)$ case in Appendix F. This formula helps estimate the irrationality measure, a critical dynamical metric for our work. Specifically, the asymptotic behavior of $\tilde{q}_n$ and $\lambda_1/\lambda_2$ are still required for finding $\delta_{\text{predicted}}$, but they are usually easier to derive.

### 3.3 Discovery of Formulas by Unsupervised Learning

Each PCF formula is defined by the polynomials that generate it. This work focuses on polynomials up to 2nd degree: $a_n = A_2 n^2 + A_1 n + A_0$, $b_n = B_2 n^2 + B_1 n + B_0$, with integer coefficients in the domain $-5 \leq A_i \leq 5$, $-5 \leq B_i \leq 5$. We removed the $a = 0$ and $b = 0$ cases, as they break the PCF structure, leaving us with 1,768,900 formulas. Some of these PCFs do not converge to a single limit, rendering their measured metrics meaningless (see Appendix B for the classification method we developed to predict PCF convergence). We filtered out all formulas that do not converge, providing the final filtered dataset of 1,543,926 formulas.

The conventional classification of the PCFs is by the coefficients of their polynomials $a_n, b_n$, $(A_2, A_1, A_0, B_2, B_1, B_0)$, and by their numerical limit $L$, which we evaluate here at depth $n = 2000$.

Going beyond these conventional classification, our methodology relies on dynamics-based metrics calculated for each formula:

- The irrationality measure: for each PCF, we calculate $\delta_{\text{predicted}}$ (Eq.5) and compute $\delta$ directly using the Blind-$\delta$ algorithm (presented in section 3.4) at depth $n = 1000$. Fig.2a presents example $\delta$ evaluations.

- The convergence rate dynamics, comprised of three parameters: for each PCF, we fit the approximation error $\epsilon(n) := \left| \frac{p_n}{q_n} - L \right|$, which scales as $\epsilon(n) \sim n!^{\eta} \cdot e^{\gamma n} \cdot n^{\beta}$ for large $n$. We store the fitted $\eta$ (factorial coefficient), $\gamma$ (exponential coefficient), and $\beta$ (polynomial coefficient). The process is detailed in Appendix A.

- The growth rate of $\tilde{q}_n$, comprised of three parameters: For each PCF, we fit the denominator to $\tilde{q}_n \sim n!^{\eta'} \cdot e^{\gamma' n} \cdot n^{\beta'}$ for large $n$. We store the fitted parameters $\eta', \gamma', \beta'$.

Based on this set of metrics, we applied unsupervised clustering for unlabeled data (using the hierarchical density-based HDBSCAN algorithm [McInnes et al., 2017]). The clustering is the key component in the algorithm we developed (Fig.1), leading to the discovery of a variety of formulas and data patterns that exposed formula families (see sections 4.1, 4.2, 4.3, and 4.4 for selected results).

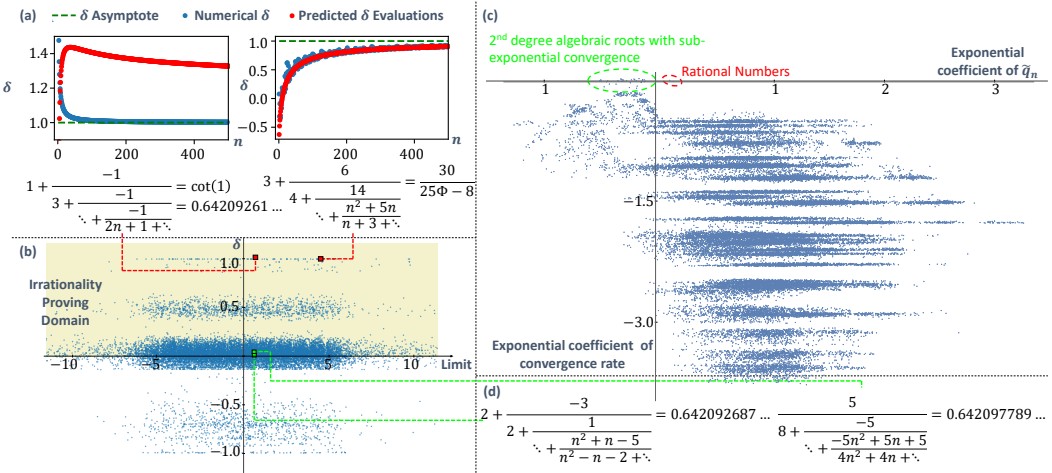

Figure 2: **Dynamics-based metrics for formulas of mathematical constants.** Analysing the convergence of polynomial continued fraction (PCF) formulas provide *dynamical metrics* that prove useful for their automated clustering and identification. (a) Irrationality measure $\delta$ vs. PCF depth, for two example formulas of the constants $\cot(1)$ and the Silver Ratio. The $\delta$ of these constants is known to be 1 (green dashed lines). The blue dots show the numerical convergence of $\delta$ (Eq.3) to the correct value. The red dots show the evaluated $\delta$-Predictor formula (Eq.5), following the numerical $\delta$ very closely in the Silver Ratio formula, while taking a completely different (and much slower) trajectory in the $\cot(1)$ formula; yet both converge to the correct value $\delta = 1$. For the purposes of clustering, $\delta_{\text{predicted}}$ was evaluated at $n = 10^9$, providing an accurate estimation for $\delta$. (b) $\delta_n$ ($n = 1000$) vs. the limit value for PCFs in our dataset. While $\delta$ values seem to follow a pattern, the limit value distribution does not contain relevant information (the higher density of PCFs near the Y axis arises from the small coefficients of the polynomials in our dataset). Our dataset contains $913,056$ irrationality-proving formulas, most of which are not yet linked to any known constant. (c) Exponential growth coefficients of $\tilde{q}_n$ and of $\epsilon(n)$ for PCFs with $\deg(B) = 2\deg(A)$. Note the surprising "band-structures" that this view reveals. A few of the clusters have been identified, but the reason for the appearance of these "bands" and the properties of most clusters remain as open questions. (d) Example PCFs in the dataset that converge to a value close to the constant $\cot(1)$ ($\pm 10^{-5}$) and yet are not related to it, showcasing the challenge of mathematical formula discovery. For visual clarity, error bars not shown. See Appendix A for a discussion regarding measurement errors.

## 3.4 The Blind-$\delta$ Algorithm

The irrationality measure $\delta$ of a PCF is of mathematical interest, and (as we will see in section 4) is a powerful dynamical metric. Unfortunately, evaluating $\delta$ using Eq.3 requires knowing the series limit $L$, making its estimation for a large set of unlabeled PCFs impractical.

The Blind-$\delta$ algorithm was created in order to circumvent this limitation. Instead of inspecting the convergence behavior of $\frac{p_n}{q_n} \to L$, we inspect the convergence behavior of $\frac{p_n}{q_n} \to \frac{p_m}{q_m}$ for specific

$m > n$. Given a rational approximation $\frac{p_n}{q_n} \to L$, we approximate the error rate $\epsilon(n)$ with

$$\left| \frac{p_n}{q_n} - \frac{p_m}{q_m} \right| = \epsilon(n) \cdot \left| 1 - \frac{\epsilon(m)}{\epsilon(n)} \right|.$$

If $0 < s < \left| 1 - \frac{\epsilon(m)}{\epsilon(n)} \right| < S$ is bounded away from zero and infinity for all $n$ large enough, then the approximation of the Blind-$\delta$ algorithm has the same convergence rate as Eq.3, bypassing the need to evaluate $L$. Intuitively, this condition holds whenever $|\epsilon(n)| \to 0$ fast enough, which is true for the vast majority of PCFs (see Appendix F for details).

Note that $m$ has to grow with $n$. In practice, our implementation of the algorithm in this work uses $m = 2n$, so in order to study $\delta$ up to $n = 1000$, we use $m = 2000$.

### 3.5 Choice of Metrics for Clustering

As part of the automated formula discovery flow we choose the best metric (for each step), in terms of representation power, which is measured by applying the Davies-Bouldin Index [Davies and Bouldin, 1979] on clustering using each metric individually. Table 1 shows results for a randomly chosen sample of 25K converging PCFs. Note the extremely poor performance of the PCF limit $L$, in agreement with Fig.2b,d. This dimensionality reduction is important for efficiency during the clustering step and for better explainability. The former is because the dataset size grows exponentially with the PCF degree and with the magnitude of the polynomial coefficients.

Table 1: Comparison of the representation power of the main dynamical metrics (lower is better). $\beta$, $\beta'$ and $(A_2, A_1, A_0, B_2, B_1, B_0)$ provide little value for the initial clustering and are not shown.

| Metric | | Davies-Bouldin Index |
|---|---|---|
| Limit $L$ | | 67.23 |
| Irrationality measure $\delta$ | | 1.11 |
| Reduced denominator $\tilde{q}_n$ growth factors $\tilde{q}_n \sim n!^{\eta'} \cdot e^{\gamma' n} \cdot n^{\beta'}$ | Exponential coefficient $\gamma'$ | 0.51 |
| | Factorial coefficient $\eta'$ | 0.13 |
| Error rate $|e(n)|$ growth factors $|e(n)| \sim n!^{\eta} \cdot e^{\gamma n} \cdot n^{\beta}$ | Exponential coefficient $\gamma$ | 14.83 |
| | Factorial coefficient $\eta$ | 0.77 |

Other metrics were tested. Some have been shown to have little to no representation power (e.g. $p_n$ and $q_n$, as defined in Eq.2, modulo various primes, their sign, their GCD etc.) while others show potential and are left for future study (e.g. the leading Fourier coefficients of the "noise" around the fit of $\tilde{q}_n$). A relatively small number of metrics were measured and used, which helped keep the results mathematically explainable. Nevertheless, the clustering using the metrics in Table 1 showcases the strength of our dynamical metrics approach.

## 4 Results

### 4.1 Discovered Formulas for Mathematical Constants

The first step in validating the dynamical metrics approach is using basic heuristics on the metric space to find PCFs related to mathematical constants. There are some PCFs in the dataset that have a known irrational limit (like the examples in Eq.1 and the PCF family

$$\cfrac{B}{A + \cfrac{B}{A + \ddots}} = \frac{2B}{A + \sqrt{A^2 + 4B}}$$

for constant $A$ and $B$), so we expected to find some of them. Through this test, we also found *previously unknown* PCF formulas related to mathematical constants. Note that known mathematical formulas are both the anchors for labeling and a test set in our method.

A natural heuristic is inspecting PCFs with $\delta \approx 1$, marking their limits as irrational. Another heuristic we used is focusing on PCFs with $\eta' \approx 0$, as it was a very strong indicator for mathematical constant

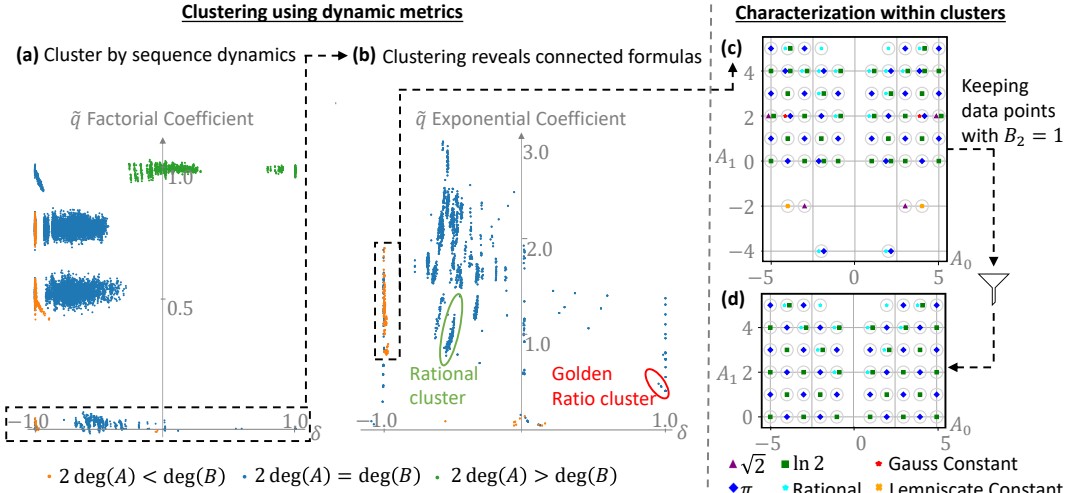

**Figure 3: Discovery of mathematical structures via analysis of dynamical metrics of formulas.**
(a) Projecting the data on the $\delta$ vs. $\eta'$ ($\tilde{q}_n$ factorial coefficient) plane, it is easy to see the emerging subsets. We focus on PCFs with $\eta' \approx 0$, as a previous work [Elimelech et al., 2023] indicated this as an important property. (b) Clustering in the $\delta$ vs. $\gamma'$ ($\tilde{q}_n$ exponential coefficient) plane shows examples of common properties within a cluster, like rationality or convergence to a specific constant (up to a linear fractional transformation). Focusing further on the $\deg(B) > 2\deg(A)$ cluster (as it is a clear anomaly in the $\eta' \approx 0$ subset), we used a PSLQ algorithm to identify links between these formulas and mathematical constants. This identification was feasible since a preliminary step identified a promising subset $\sim 5,000$ times smaller than the initial dataset. (c) The result of this clustering and identification procedure is a structured arrangement of formulas that reveal a range of novel formulas related to constants such as $\pi$, $\ln(2)$, $\sqrt{2}$, Gauss' constant, and Lemniscate's constant. (d) Keeping only PCFs with $B_2 = 1$ we are left with a highly symmetrical "checkerboard pattern" of formulas for $\pi$ and $\ln(2)$, which was generalized into infinite formula families hypotheses (see section 4.3). Error bars not shown for visual clarity, see Appendix A for a discussion regarding measurement errors.

formulas in a previous work [Elimelech et al., 2023]. Combining the two gives a subset (see Fig.3a top left) that contains PCFs such as:

$$5 + \cfrac{-10}{\ddots + \cfrac{-5n^2 - 5n}{5n + 5 + \ddots}} = 2 + \phi \qquad -3 + \cfrac{1}{\ddots + \cfrac{1}{-3 + \ddots}} = \frac{-2}{\sqrt{13} - 3} \tag{6}$$

Removing the limitation on $\tilde{q}_n$ growth rate, one can find the $\cot(1)$ formula shown in Fig.2a:

$$1 + \cfrac{-1}{\ddots + \cfrac{-1}{2n + 1 + \ddots}} = \cot(1) \tag{7}$$

On the other hand, relaxing the limitation on $\delta$, focusing only on $\eta' \approx 0$, a rich structure emerges (Fig.3b). Diving deeper into the B-dominated subset, we find formulas (Fig.3c) for the Gauss constant $G_{GA}$ [Finch, 2003]:

$$4 + \cfrac{6}{\ddots + \cfrac{4n^2 + 2n}{4 + \ddots}} = \frac{2G_{GA}}{4G_{GA} - 3} \qquad 4 + \cfrac{4}{\ddots + \cfrac{4n^2 + 2n - 2}{4 + \ddots}} = \frac{4G_{GA} - 1}{3G_{GA} - 2} \tag{8}$$

Lemniscate constant $L_{Lemniscate}$ [Finch, 2003]:

$$4 + \cfrac{2}{\ddots + \cfrac{4n^2 - 2n}{4 + \ddots}} = \frac{-6}{L_{Lemniscate} - 4} \tag{9}$$

As well as for second order roots, $\pi$ and $\ln(2)$ (see section 4.3). Note that unlike the formulas in Eq.6 and Eq.7, which are analytically proven, the formulas in Eq.8 and Eq.9 are (to the best of the authors' knowledge) *novel*. Their limits were numerically validated to a large precision, yet formal proofs for these formula hypotheses remain an open challenge.

It should be noted that usually in number theory, a bigger $\delta$ is considered "good", whereas a smaller (often negative) $\delta$ is considered "bad". We use $\delta$ as a metric, without "judgment". These novel formulas (Eq.8, Eq.9 and the infinite family of formulas shown in section 4.3), which have the "bad" $\delta \approx -1$, are a demonstration that our "non-judgmental" approach is successful.

## 4.2 Clustering in Dynamics-Based Metric Latent Space

This section shows that clusters in the latent space of dynamics-based metrics are successful in grouping together different formulas in a way that exposes their shared properties, such as the mathematical constant to which they relate.

Looking at the top left cluster in Fig.3b (defined by $\tilde{q}_n$ exponential coefficient $< 0.6$ and $\delta > 0.9$), we recognize the canonical form of the golden ratio PCF (shown in Eq.1). This cluster also contains 21 additional PCFs, with different generating polynomials, some of higher degree. As it turns out, all of them are linear fractional transformations of $\sqrt{5}$ (see Appendix C), which were labeled automatically by the formula discovery algorithm (Fig.1). Another example of property conservation within clusters is the rational cluster marked in green on Fig.3b. The limits of the PCFs in this subset are varied, and its spread is real (i.e., not only due to numerical imperfections). Yet, all the PCFs in this cluster converge to rationals - which is not directly measured by any of the latent space dimensions.

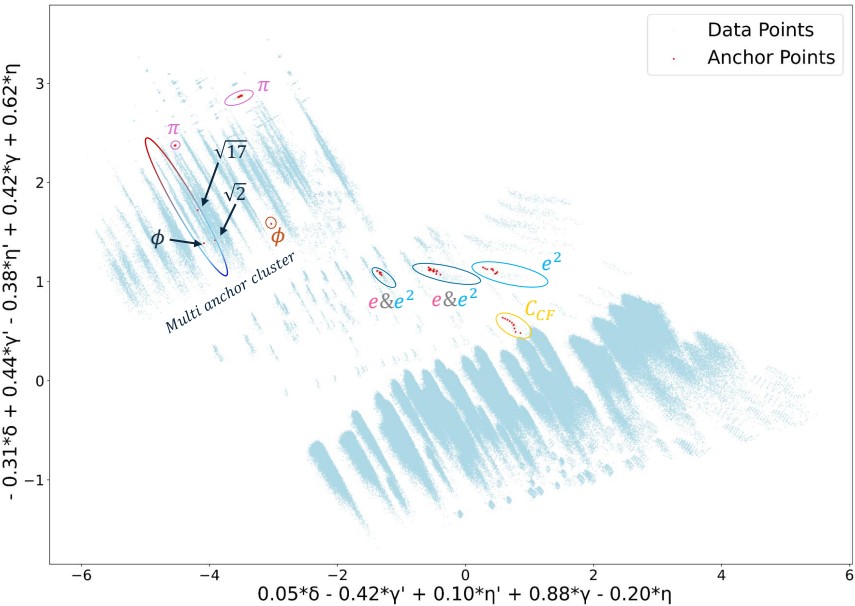

Figure 4: **Automated Formula Discovery Results:** Showcasing the automated clustering and labeling of PCFs using a set of 126 known anchor formulas, connected to constants such as $\pi, e, e^2$, $C_{CF}$ (the continued fraction constant), the golden ratio $\phi, \sqrt{2}, \sqrt{3}$, and $\sqrt{17}$. The clustering is visualized here via the 2 leading PCA components, revealing *441 novel, automatically discovered, mathematical formula hypotheses*. For visual clarity, error bars are not shown. See Appendix A for a discussion regarding measurement errors. See Appendix E for additional visualizations.

Fig.4 showcases a collection of clusters with shared properties. Using a set of 126 (mathematically unique) known anchor formulas, *441 novel mathematical formula hypotheses were automatically*

*discovered*. The constants which are related to the most new conjectures are: $e^2$ (28 anchor formulas gave 178 new conjectures), $\pi$ (39 anchor formulas gave 116 new conjectures), $e$ (44 anchor formulas gave 80 new conjectures), and $\sqrt{17}$ (1 anchor formula gave 55 new conjectures). Some of the novel formulas are equivalent to known PCFs (see Appendix C for a discussion about equivalence) while other formulas were analytically proven (see Appendix A.3 and Appendix G).

Note the multi-anchor clusters of $e$ and $e^2$, as well as the algebraic roots: these clusters failed to single out a specific constant, yet relate to constants of similar nature - suggesting meaningful clustering nevertheless. For the sake of visualization the algorithm stopped after the second iteration. In a standard run these multi-anchor clusters would have been separated via additional metrics.

### 4.3 Detecting Patterns and Underlying Structure

As mentioned in section 4.1, the $\deg(B) > 2\deg(A)$, $\eta' \approx 0$ cluster, contains many formulas of interest (see Fig.3c and d). They were discovered via a PSLQ algorithm, identifying linear fractional relations between the limit values of PCFs in the subset and notable mathematical constants (such as $\pi$ or $e$). This is a computationally heavy operation, and it would be challenging to run it on all 1.5M formulas in the data set. Yet by first identifying the promising clusters, we reduce the search space $\sim 5,000$ times, allowing for a deeper inspection of each PCF.

Once the "checkerboard" pattern in Fig.3d was discovered, the hypothesis was expanded into 2 infinite families of PCFs with sub-exponential convergence relating to $\pi$ and $\ln(2)$:

- $a_n = i + 2j + 1, b_n = n^2 + (i + k)n$, with integers $i, j \geq 0$, and $k \in \{0, 1\}$. This is expected to be related to $\pi$ if $k = 1$, and to $\ln(2)$ if $k = 0$ (in fact, this pattern can be generalized even further, into a novel 3-dimensional Conservative Matrix Field, provided in Appendix C. See [Elimelech et al., 2023] for the definition of Conservative Matrix Fields).

Another formula family was discovered via clustering in the $\gamma$ vs. $\gamma'$ space. The algebraic roots subset (marked by a green circle in Fig.2c) was generalized into:

- $a_n = -2n + j - 1, b_n = -n^2 + jn + k$ for integer $j, k$ such that $b_n$ has real roots that are not positive integers. This is expected to converge to a root of $b_n$.

These are novel experimental results and mathematical hypotheses - awaiting proof.

### 4.4 Higher Degree PCF Identification

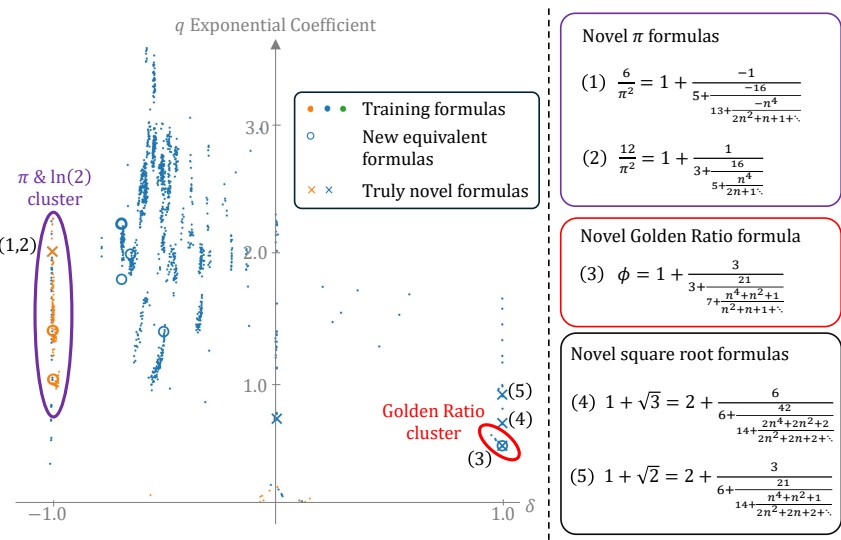

Figure 5: Higher-degree PCF formulas for mathematical constants, overlaid on Fig.3b. Formulas that are equivalent to an existing element in the original dataset are marked with "o", while "x" marks truly novel formulas. *The higher-degree PCFs fit, using the same dynamical metrics, into clusters trained on lower-degree PCFs, showing the ability to identify novel formulas despite being of mathematical forms not seen during training.*

The final validation for the dynamical metrics approach is to show its potential even on test sets of a different mathematical structure than the training set. For that purpose, the clusters created based on the original dataset were treated as a classifier, and a test set of higher-degree PCFs (up to $3^{rd}$ degree $A_n$ and $4^{th}$ degree $B_n$, coefficient range [-2,2]) was created, measured and classified.

Naturally, not all high-degree PCFs fit neatly into the existing clusters (as they represent constants that were not present in the original dataset), but some were correctly identified and labeled, discovering novel formulas in the process (see Fig.5 for a sample of the results).

## 5  Discussion and Outlook

This work marks an important step toward the vision of automated on-demand formula creation in mathematics. Going beyond all previous algorithms in this field, we connect the challenge of formula creation to robust ML methods. This methodology provides a wide variety of automatically generated formulas, including both previously known and previously unknown ones, exposing their underlying mathematical structure and enabling new proofs.

The next research step directly building on our methodology could help to finally reveal the complete intricate mathematical structure of PCFs. For example, starting with the "band-structure" found in Fig.2c, or with clusters of formulas with various structures representing the same mathematical constant. Further exploration of our conjectures from section 4 could have more impact on mathematics, perhaps achieving further generalizations and prescriptive formula generation.

The technique presented here can be applied to a larger scope of continued fractions and for completely different types of formulas. For more general continued fractions, dynamical metrics such as the numerical trajectories and the corresponding sequences of $\delta$ (in addition to its asymptotic value) hold valuable information even in continued fractions that do not converge at all. We expect these dynamical metrics to provide a "fingerprint" for wider families of formulas and perhaps even for the mathematical constants themselves. This approach was directly applied in this work to higher polynomial degrees, larger polynomial coefficients, and can be expanded to continued fractions not based on polynomials. Looking beyond continued fractions, metrics that are derived from the dynamics of a numerical calculation of certain formulas are an especially good fit for automated computer-assisted investigations. Such metrics can be measured for a variety of mathematical structures, including ones whose evaluation is iterative or recursive, that are defined via an infinite sum, or any other process which produces rational approximants. Any such mathematical structure can be measured, clustered, and identified using the proposed method - treating the generating functions as a black box. We believe that such dynamical metrics can unveil patterns and underlying structures in broad fields of mathematics and in other areas of science.

To exemplify this universal concept, we looked into higher depth recursion relations, which are a promising research direction because little is known about their global structure, yet they are involved in several important conjectures. For example, the best rational approximation formula known for Euler's gamma constant is constructed via such a recursion relation [Aptekarev, 2009]. This family of formulas is broader than continued fractions, yet they can be described by the same metrics as PCFs. Another type of mathematical structure successfully analyzed using the same method is hypergeometric functions, showing the applicability of our measurement-clustering-generation approach to a bigger family of mathematical functions. This generalization can be useful in a wide variety of contexts, such as investigations of integral formulas (e.g., Beukers-type integrals [Beukers, 1979, Dougherty-Bliss et al., 2022, Brown and Zudilin, 2022]).

Our work was based on a limited-size dataset and on a small set of metrics. It would be intriguing to test the extracted conjectures on larger datasets, which can help reveal additional, more intricate, phenomena. Considering the success we had using a relatively small set of metrics, we would like to use an order-of-magnitude larger set of metrics and find what new predictions can be recovered. In fact, the creation and evaluation of the metrics themselves can be automated.

Taking a broader perspective, the methodology presented in this work can be seen as a general prescription for tackling scientific discovery challenges in mathematics and theoretical physics that rely on numerical evaluations and generalizations. Such an advance is especially exciting for such challenges that were considered in the past to require intuitive leaps of creativity.

**Acknowledgments**

This research received support through Schmidt Sciences, LLC.

# 6 References

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

# A  Numerical Measurements, Curve Fitting and Validation of Formulas

## A.1  Numerical Measurements

When characterizing PCFs, we use several metrics extracted from the dynamic behavior of the formula:

- The growth coefficients $\eta, \gamma, \beta$ (of the form $n!^\eta \cdot e^{\gamma n} \cdot n^\beta$) of the convergence rate $\epsilon(n)$.

- $\tilde{q}_n$ (as defined in Eq.3) growth coefficients: $\eta', \gamma'$ (of the form $n!^{\eta'} \cdot e^{\gamma' n}$).

- The $\delta$ (as defined in Eq.3) calculated using the Blind-$\delta$ Algorithm described in section 3.4.

To measure the growth coefficients of $\tilde{q}_n$ and $\epsilon(n)$, the values of $\log\left(\epsilon(n)\right)$ (see section 3.4) and of $\log\left(\tilde{q}_n\right)$ were evaluated up to depth 1000.

The most resource-intensive values that are generated are $p_n, q_n$ and $\gcd(p_n, q_n)$ - all other values are calculated from them (and require less precision). For the worst case PCF this requires 36MB of memory (without optimizations) and $\sim 1.9$ seconds of run time on a single core of a basic workstation, which translates to an upper cap of $\sim 900$ hours for the whole data set. In practice we used a high power cluster with 64 cores, which ran each iteration of the measurements in $\sim 8.5$ hours.

Once these values are calculated, using scipy [Virtanen et al., 2020] and numpy [Harris et al., 2020] a fit of the form $\log\left(n!^\eta \cdot e^{\gamma n} \cdot n^\beta\right)$ was calculated for $\tilde{q}_n$ and $\epsilon(n)$, producing the dynamical metrics.

## A.2  Curve Fitting

A curve fit using 1000 points is a fairly heavy operation, unsuited for large scale investigations. Instead, we used an extreme down-sampling. Specifically, only 5 points were used for the fit.

One may justifiably wonder if 5 data points are sufficient to fit accurately enough the desired metrics.

A test comparing between a 5 data point fit and a 1000 data point fit was done. As the test set, 50 PCFs were randomly chosen out of each of 9 categories (450 total test cases). The categories were all combinations of $\deg(a) = 0, 1, 2$ and $\deg(b) = 0, 1, 2$. Focusing on the dominant coefficients ($\gamma$ and $\eta$), for each case, a full (1000 point) fit was performed (producing $\gamma_f, \eta_f$), and compared to the down sampled fit of 5 points (producing $\gamma_p, \eta_p$). We tested 2 methods of choosing the 5 points, even ($i = 6, 206, 406, 606, 806$) and logarithmic ($i = 6, 125, 250, 500, 1000$). The relative error was then calculated ($\frac{|\gamma_p - \gamma_f|}{|\gamma_f|}$ and $\frac{|\eta_p - \eta_f|}{|\eta_f|}$) for $\epsilon(n)$ and $\tilde{q}_n$. The relative errors were then averaged over the test set (results summarised in table 2) - showing the 5-point fit to be almost as good as the full 1000-point fit. In our measurements we use the logarithmic point distribution as it gives better results for most metrics.

Table 2: Comparison between 1000 point fit results and 5 point fits results (even spread and logarithmic spread).

| Behavior | $\tilde{q}_n$ | | Convergence Rate | |
|---|---|---|---|---|
| Coefficient | $\gamma'$ | $\eta'$ | $\gamma$ | $\eta$ |
| Relative Error Average (even spread) | 0.0124 | 0.0007 | 0.0078 | 0.0005 |
| Relative Error Average (logarithmic spread) | 0.0175 | 0.0004 | 0.0024 | 0.00012 |

Licences and versions of Python packages (used for curve fitting, clustering and large number mathematics):

Scipy (Version: 1.11.3) - BSD License (Copyright (c) 2001-2002 Enthought, Inc. 2003-2024, SciPy Developers. All rights reserved.)

gmpy2 (Version: 2.1.5) - GNU Lesser General Public License v3 or later

Numpy (Version 1.26.1)- BSD License (Copyright (c) 2005-2023, NumPy Developers. All rights reserved.)

### A.3 Validation of Automatically Generated Formulas

After a PCF has been identified as potentially connected to a known constant $C$, it undergoes a verification process in order to establish whether it indeed converges to a known mathematical expression involving $C$. This process consists of several steps:

*1. Expression Identification Through PSLQ Analysis*

The PCF limit is computed and a targeted PSLQ search is used to identify an expression that matches the resulting digits and incorporates the constant $C$.

*2. Expression Verification*

After a candidate expression has been identified, the PCF is computed to a greater depth. The resulting values are compared against the proposed expression to assess the accuracy of the match and verify convergence.

*3. Systematic Proof*

Once the expression has been verified, an automated analytical proof is attempted via the following steps.

Given two polynomials $a(n)$ and $b(n)$ defining a PCF, Euler's formula [Euler, 1748] states that when there exist polynomials $h_1(n), h_2(n), f(n)$ such that $b(n) = -h_1(n)h_2(n)$, $f(n)a(n) = f(n-1)h_1(n) + f(n+1)h_2(n+1)$, the limit of the PCF equals:

$$\frac{f(-1)h_1(0)}{f(0)} + \frac{f(1)h_2(1)}{f(0)} \cdot \left( \sum_{k=0}^{\infty} \frac{f(0)f(1)}{f(k)f(k+1)} \prod_{i=1}^{k} \frac{h_1(i)}{h_2(i+1)} \right)^{-1} \tag{10}$$

The proof is then completed using known identities of infinite sums.

If the PCF's polynomials do not satisfy Euler's formula's conditions, the PCF is flagged for further examination.

For example, one may take the PCF defined by the polynomials $a_n = -2$ and $b_n = 4n^2 + 4n + 1$ which was labeled as related to $\pi$. The PCF was numerically identified and validated as converging to $1 - \frac{3}{3-\frac{3\pi}{4}}$ Next, the polynomials $h_1, h_2$ and $f$ were identified as $2n+1, -(2n+1)$ and $1$ respectively, which corresponds to the sum $1 - \frac{3}{\sum_{k=0}^{\infty} \prod_{n=1}^{k} \frac{-2n-1}{2n+3}}$. This infinite sum is known and was indeed proven to converge to $1 - \frac{3}{3-\frac{3\pi}{4}}$.

For the entire list of proven formulas, see table 5 in appendix G.

# B    Classification of Continued Fractions

Not all PCFs converge. Clearly, if $\frac{p_n}{q_n}$ does not have a well defined limit, then some of our numerically measured metrics lose their meaning. Though we had algorithmic safeguards to detect such cases and remove them from the analyzed set, it was valuable to identify a pattern and formulate a rule-set that predicts the convergence of a PCF.

For that purpose we turned to the matrix representation of a continued fraction to depth $n$ (see Appendix F.1 for details):

$$\begin{bmatrix} p_{N-1} & p_N \\ q_{N-1} & q_N \end{bmatrix} = \prod_{n=1}^{N-1} \begin{bmatrix} 0 & b_n \\ 1 & a_n \end{bmatrix} =$$

$$\left( \prod_{n=1}^{N-1} a_{n-1} \right) \begin{bmatrix} 1 & 0 \\ 0 & \frac{1}{a_0} \end{bmatrix} \left( \Pi_{n=1}^{N-1} \begin{bmatrix} 0 & \frac{b_n}{a_n a_{n-1}} \\ 1 & 1 \end{bmatrix} \right) \begin{bmatrix} 1 & 0 \\ 0 & a_{N-1} \end{bmatrix}$$

Assuming $a_n \neq 0$ for $n \geq 0$.

Analyzing the eigenvalues of the matrices within the matrix product as $n \to \infty$ allows for examining the asymptotic behavior of the continued fraction. Their characteristic polynomial is $\lambda^2 - \lambda - \frac{b_n}{a_n a_{n-1}}$,

and we propose observing the discriminant of this polynomial - more specifically its dominant power of $n$:

$$\Delta_n = 1 + \frac{4b_n}{a_n a_{n-1}} = C_s n^s + O(n^{s-1}) \tag{11}$$

Here we assume $C_s \neq 0$ and $s$ is some integer. Based on the data, we compiled table 3 as a summary of the conjectured behavior of any polynomial continued fraction based on $s$ and $C_s$.

Table 3: Summary of PCF behavior characterized by $s$ and $C_s$ as defined in Eq.11.

| Convergence | $C_s > 0$ | $C_s < 0$ |
|---|---|---|
| ✗ | $s \geq 3$ | $s \geq 0$ |
| ✓ | $s \leq 2$ | $s \leq -1$ |

We can further elaborate on the converging cases by discussing the conjectured rate of convergence. Usually, a PCF is expected to converge at a sub-exponential rate, but in the case of $s = 0, C_0 > 0$ it is expected to converge faster:

- If $C_0 \neq 1$ then the PCF will converge at an exponential rate, and the exact rate of convergence increases monotonically as $C_0 \to 1$, with a vertical asymptote at $C_0 = 1$. The convergence rate is identical for $C_0$ and $\frac{1}{C_0}$.

- If $C_0 = 1$ then the PCF will converge at a factorial rate. More specifically, if we find the second most dominant power $\Delta_n = C_0 + \frac{C_t}{n^t} + O(\frac{1}{n^{t+1}})$ for some $C_t \neq 0$ and integer $t > 0$ then the precision will grow at a rate of $O(n!^t)$.

We used these rules (in conjunction with the measurements mentioned in section 3.3) to validate that all PCFs we analyze and cluster do converge and their measured metrics are well defined.

## C   Discovering equivalence of continued fractions

Polynomial continued fractions use two polynomials $a_n = a(n)$ and $b_n = b(n)$ to generate a sequence of rationals $p_n/q_n$. However, the same sequences with identical behaviour can be generated using more then one set of polynomials. By identifying transformations under which the dynamics of $p_n/q_n$ remains invariant, we can formally prove equivalence between data points, validating the clustering power of the chosen metrics.

By rearranging the continued fraction definition, we can see how equivalent $a_n$ and $b_n$ series can arise:

$$a_0 + \cfrac{b_1}{a_1 + \cfrac{b_2}{a_2 + \cfrac{b_3}{\ddots + \cfrac{b_n}{a_n + \ddots}}}} = a_0 \left( 1 + \cfrac{\frac{b_1}{a_0 a_1}}{1 + \cfrac{\frac{b_2}{a_1 a_2}}{1 + \cfrac{\frac{b_3}{a_2 a_3}}{\ddots + \cfrac{\frac{b_n}{a_n a_{n-1}}}{1 + \ddots}}}} \right) = \frac{a_0 c_0}{c_0} \left( 1 + \cfrac{\frac{b_1 c_0 c_1}{a_0 c_0 a_1 c_1}}{1 + \cfrac{\frac{b_2 c_1 c_2}{a_1 c_1 a_2 c_2}}{1 + \cfrac{\frac{b_3 c_2 c_3}{a_2 c_2 a_3 c_3}}{\ddots + \cfrac{\frac{b_n c_n c_{n-1}}{a_n c_n a_{n-1} c_{n-1}}}{1 + \ddots}}}} \right) \tag{12}$$

Indeed, by defining a new pair of polynomials $a_n' = a_n c_n; b_n' = b_n c_n c_{n-1}$ we get an equivalent continued fraction which converges to $\frac{c_0 p_n}{q_n}$. Clearly, since the resulting sequence $\frac{p_n'}{q_n'}$ is identical to the original one, it exhibits the same dynamics. We call this process "Inflation by $c_n$".

The metrics we are interested in are mostly not affected by a finite number of elements in the sequence. For example, both the convergence rate and $\delta$ discuss an overall trend as $n$ grows. Consequently, we can initiate the sequence at different values of $n \neq 0$ without changing the latent parameters. When expressing these transformations as modification to the continued fraction definition, we see that the *limit* of the continued fraction might change due to this shift in sequence initiation, but only by a rational fractional transform.

$$a_0 + \cfrac{b_1}{a_1 + \cfrac{b_2}{a_2 + \cfrac{b_3}{\ddots + \cfrac{b_n}{a_n + \ddots}}}} = \frac{p_n}{q_n} \Rightarrow a_1 + \cfrac{b_2}{a_2 + \cfrac{b_3}{\ddots + \cfrac{b_n}{a_n + \ddots}}} = \frac{b_1}{\frac{p_n}{q_n} - a_0}$$

For example, we consider the cluster of formulas related to the golden ratio shown in figure 3b. A large portion of these PCFs stem from transforming the known formula for the golden ratio shown in Eq.1 via the methods aforementioned. The exact transformations are depicted in Table 4.

Table 4: Continued fractions converging to linear fractional transformations of the golden ratio $\phi$, found using the top left cluster of Figure 3b. Numerous data points in this cluster exhibit identical sequence dynamics and are equivalent under the inflation and index indentation transformations. The equivalent data points create families of continued fractions in the cluster. Discrepancies between the calculated irrationality measure within the same family is ascribed to numerical inaccuracies, typically on the order $0.001$. However, when comparing families, discrepancies in the irrationality measure rise to a magnitude of $0.04$, suggesting potential deeper distinctions among these PCFs.

| $A_n$ | $B_n$ | Limit | Transformation | Irrationality measure $\delta$ |
|---|---|---|---|---|
| 1 | 1 | $\phi$ | Family's canonical form | $\delta = 1.00168$ |
| $-1$ | 1 | $-\phi$ | Inflation by $c_n = -1$ | $\delta = 1.00168$ |
| 2 | 4 | $2\phi$ | Inflation by $c_n = 2$ | $\delta = 1.00023$ |
| $-2$ | 4 | $-2\phi$ | Inflation by $c_n = -2$ | $\delta = 1.00023$ |
| $n+1$ | $n(n+1)$ | $\phi$ | Inflation by $c_n = n+1$ | $\delta = 1.00168$ |
| $-(n+1)$ | $n(n+1)$ | $-\phi$ | Inflation by $c_n = -(n+1)$ | $\delta = 1.00168$ |
| $n+2$ | $(n+1)(n+2)$ | $2\phi$ | Inflation by $c_n = n+2$ | $\delta = 1.00023$ |
| $-(n+2)$ | $(n+1)(n+2)$ | $-2\phi$ | Inflation by $c_n = -(n+2)$ | $\delta = 1.00023$ |
| $2n+1$ | $(2n-1)(2n+1)$ | $\phi$ | Inflation by $c_n = (2n+1)$ | $\delta = 1.00168$ |
| $-(2n+1)$ | $(2n-1)(2n+1)$ | $-\phi$ | Inflation by $c_n = -(2n+1)$ | $\delta = 1.00168$ |
| $2(n+1)$ | $4n(n+1)$ | $2\phi$ | Inflation by $c_n = 2(n+1)$ | $\delta = 1.00023$ |
| $-2(n+1)$ | $4n(n+1)$ | $-2\phi$ | Inflation by $c_n = -2(n+1)$ | $\delta = 1.00023$ |
| 5 | $-5$ | $\phi+2$ | Family's canonical form | $\delta = 1.00168$ |
| $-5$ | $-5$ | $-(\phi+2)$ | Inflation by $c_n = -1$ | $\delta = 1.00168$ |
| $5(n+1)$ | $-5n(n+1)$ | $\phi+2$ | Inflation by $c_n = n+1$ | $\delta = 1.00168$ |
| $-5(n+1)$ | $-5n(n+1)+0$ | $-(\phi+2)$ | Inflation $c_n = -(n+1)$ | $\delta = 1.00168$ |
| $n+2$ | $n(n+3)$ | $(30\phi+6)/19$ | Family's canonical form | $\delta = 0.96967$ |
| $-(n+2)$ | $n(n+3)$ | $-(30\phi+2)/19$ | Inflation by $c_n = -1$ | $\delta = 0.96967$ |
| $n+3$ | $(n+1)(n+4)$ | $(30\phi+2)/11$ | Indent $n \rightarrow n+1$ | $\delta = 0.97245$ |
| $-(n+3)$ | $(n+1)(n+4)$ | $-(30\phi+2)/11$ | Indent $n \rightarrow n+1$ and inflation by $c_n = -1$ | $\delta = 0.97245$ |
| $n+3$ | $n(n+5)$ | $(750\phi+240)/361$ | Family's canonical form | $\delta = 0.95243$ |
| $-(n+3)$ | $n(n+5)$ | $-(750\phi+240)/361$ | Inflation by $c_n = -1$ | $\delta = 0.95243$ |

# D    Scalability to Larger Datasets

The methodology can be scaled to larger data sets in multiple ways. For example, by extending the range of PCF coefficients from [-5, 5] to [-10, 10], the size of the data set increases by approximately 50 times. Additionally, by considering polynomials of higher degrees - such as advancing to third-degree $a_n$ and fourth degree $b_n$ with coefficients in [-5, 5] - the dataset size can be amplified by approximately 1,333 times.

To manage the substantial computational demands associated with these expansions, a three-fold solution is proposed.

First, a dynamically chosen measurement depth is implemented during evaluation, aiming for a fixed precision across all PCFs rather than maintaining a constant depth for all computations. This approach optimizes computational efficiency by adjusting the measurement depth based on the specific convergence rate of each PCF.

Second, using known equivalences of PCFs such as inflation (see Appendix C), it is possible to substantially decrease the effective size of the dataset that needs to be measured. In particular, when $c_n = -1$, we observe that the sign of $a_n$ does not affect the dynamics of the sequence - only flips the

sign of the limit to $-L$. For every PCF its inflation by -1 is also contained in the current data set, and clearly will have the same dynamics-based metrics. This equivalence single handedly de-facto cuts the size of the current data set by half (to 771,963 converging formulas).

Third, the inherently parallelizable nature of computing metrics for each formula is leveraged. The algorithm has been adapted and deployed within the Berkeley Open Infrastructure for Network Computing [Anderson, 2004], enabling parallel computation across thousands of volunteer computers. Assuming a typical contribution of approximately 1,000 BOINC volunteer cores, processing the expanded data set requires about one month of computational time.

The approaches described enable the methodology to handle larger datasets efficiently, facilitating the analysis of more complex polynomial continued fractions within practical computational limits.

## E  Clustering Visualizations

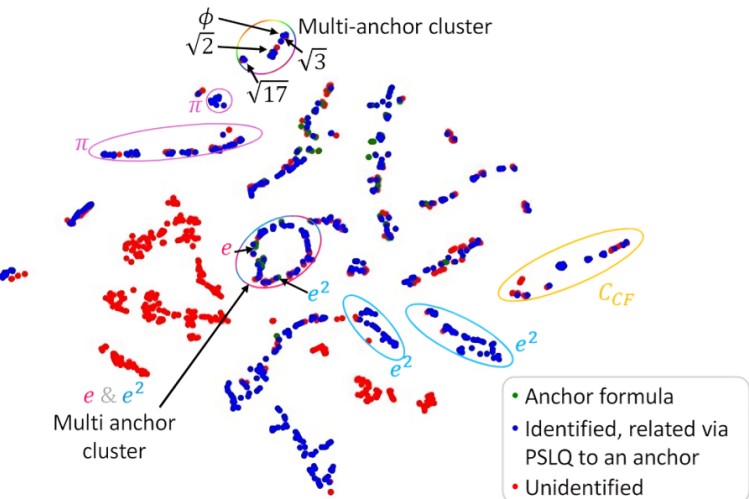

Figure 6: Automated clustering and labeling of PCFs via a 2D tSNE with perplexity $= 10$. Clusters were verified via PSLQ relations between members of the cluster and known formulas. For visual clarity not all points are shown and error bars are not shown, see Appendix A for a discussion regarding measurement errors.

Fig.6 and Fig.7 show alternate clustering and 2D visualization approaches (in addition to Fig.4). The same set of anchors was used for all 3 versions. The tSNE algorithm (Fig. 6) provides a visualization that separates well between clusters, but loses out on explainability. In Fig.7 the opposite approach was taken - using only 2 dynamical metrics for clustering allows each cluster to be defined very clearly, but information from other dynamical metrics (which can be used for better cluster separation) is lost along the way.

Despite their differences, all 3 techniques point to similar results and conclusions, providing additional validation for our conclusions.

## F  Analysis of the convergence rate

The growth rate for simple continued fraction or equivalently for constant linear recurrences is well understood, and usually boils down to the matrix defining the recurrence, and its eigenvalues. In our case, the coefficient in the recurrence also depend on $n$, so their study is more involved, however the ideas are similar, which we now describe

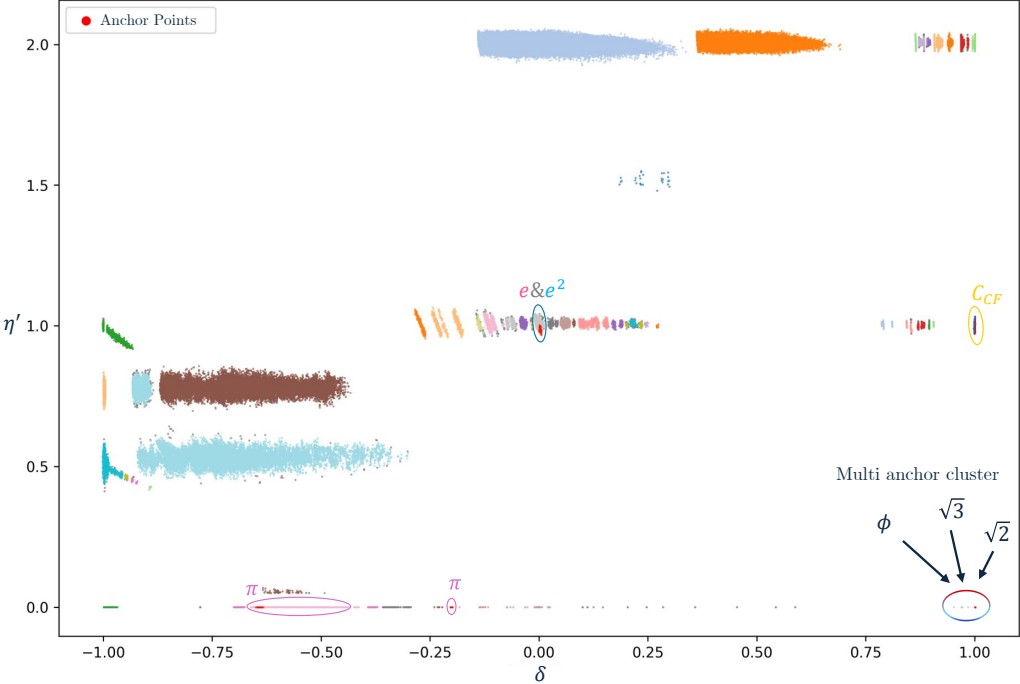

Figure 7: Automated clustering and labeling of PCFs via HDBSCAN. Only 2 dynamical metrics were used, $\delta$ and $\eta'$ (as in Fig.3a), and yet the clustering is already informative. For visual clarity error bars are not shown, see Appendix A for a discussion regarding measurement errors.

### F.1 Approximating the error rate

To find whether or not the sequence $\frac{p_n}{q_n}$ converges and if so what is its convergence rate, we note the continued fraction formula

$$\cfrac{b(1)}{a(1) + \cfrac{b(2)}{a(2) + \cfrac{b(3)}{\ddots + \frac{b(n-1)}{a(n-1)+0}}}} = \frac{p_n}{q_n},$$

can be rewritten in matrix form as

$$\begin{pmatrix} p_{n-1} & p_n \\ q_{n-1} & q_n \end{pmatrix} = \prod_1^{n-1} \begin{pmatrix} 0 & b(k) \\ 1 & a(k) \end{pmatrix}.$$

In particular this implies that both $p_n$ and $q_n$ satisfy the same linear recurrence:

$$u_{n+1} = a(n) u_n + b(n) u_{n-1},$$

with initial conditions

$$\begin{pmatrix} p_0 & p_1 \\ q_0 & q_1 \end{pmatrix} = \begin{pmatrix} 1 & 0 \\ 0 & 1 \end{pmatrix}.$$

Trying to determine if there is convergence, we use the Cauchy condition. For any $m \geq n$ we have that

$$\frac{p_m}{q_m} - \frac{p_n}{q_n} = \sum_n^{m-1} \left( \frac{p_{k+1}}{q_{k+1}} - \frac{p_k}{q_k} \right) = \sum_n^{m-1} \frac{p_{k+1}q_k - q_{k+1}p_k}{q_k q_{k+1}} = -\sum_n^{m-1} \frac{\det \begin{pmatrix} p_k & p_{k+1} \\ q_k & q_{k+1} \end{pmatrix}}{q_k q_{k+1}} = -\sum_n^{m-1} \frac{(-1)^k \prod_{j=1}^k b(j)}{q_k q_{k+1}}.$$

**Corollary 1.** The sequence $\mathbb{K}_1^\infty \frac{b(n)}{a(n)}$ converges if and only if $\sum_1^\infty \frac{\prod_{j=1}^k b(j)}{q_k q_{k+1}}$ converges, and to the same limit $L$. More over, the convergence rate is

$$\epsilon(n) := \left| \frac{p_n}{q_n} - L \right| = \left| \sum_n^\infty \frac{(-1)^k \prod_{j=1}^k b(j)}{q_k q_{k+1}} \right|.$$

This suggests that we should understand the growth rate of both $q_k$ and $\prod_{j=1}^k b(j)$.

**Remark 1.** Note that the convergence and its rate might depend on the sign of $\frac{(-1)^k \prod_{j=1}^k b(j)}{q_k q_{k+1}}$.

1. Suppose that $\left| \frac{(-1)^k \prod_{j=1}^k b(j)}{q_k q_{k+1}} \right| = \frac{1}{k^d}$. If the signs do not alternate, then $\left| \sum_n^\infty \frac{(-1)^k \prod_{j=1}^k b(j)}{q_k q_{k+1}} \right| = \sum_n^\infty \frac{1}{k^d}$. This diverge if $d = 1$ and has order of magnitude $\frac{1}{k^{d-1}}$ for $d > 1$. However, with alternating signs we get the smaller bound

$$\sum_{2n}^\infty \frac{(-1)^k}{k^d} = \sum_n^\infty \left( \frac{1}{(2k)^d} - \frac{1}{(2k+1)^d} \right) = \sum_n^\infty \left( \frac{(2k+1)^d - (2k)^d}{(2k)^d (2k+1)^d} \right) \sim \sum_n^\infty \frac{d(2k)^{d-1}}{4k^{2d}} \sim \frac{1}{n^d}.$$

   Thus, it always converges and with better rates.

2. However, for faster converging sequences we do not expect alternating sign to affect the convergence rate. For example, if $\left| \frac{\prod_1^{m-1}(-b(k))}{q_m q_{m-1}} \right| = \lambda^m$ for some $0 < \lambda < 1$, then with only positive signs the limit will be $\frac{\lambda^n}{1+\lambda}$ while for alternating signs it will be $\frac{(-\lambda)^n}{1+\lambda}$, so in any case the convergence rate is exponential.

## F.2   Growth rate of $\prod_{k=1}^{m-1} |b(k)|$

**Claim 1.** Let $b(x)$ be a polynomial of degree $d$, with leading coefficient of absolute value $B$. Then there exists a constant $C > 0$ such that for any integer $N$ we have

$$(Ne)^{-C} \leq \prod_{k=1}^N \left| \frac{b(k)}{Bk^d} \right| \leq (Ne)^C.$$

**Proof 1.** We may assume that the leading coefficient of $b$ is positive. Writing $b(x) = \sum_0^d b_j x^j$ with $b_d = B \neq 0$, we want to approximate the product (of the absolute value) of

$$\tilde{b}(k) = 1 + \sum_0^{d-1} \frac{b_j}{B} \frac{1}{k^{d-j}}.$$

Hence, we can find an integer constant $C_0 \geq 1$ such that for all $k \geq 1$ we have

$$\left( 1 - \frac{C_0}{k} \right) \leq \left| \tilde{b}(k) \right| \leq \left( 1 + \frac{C_0}{k} \right).$$

For all $k$ large enough, all the expression above are positive, so we get

$$\ln \left( 1 - \frac{C_0}{k} \right) \leq \ln \left| \tilde{b}(k) \right| \leq \ln \left( 1 + \frac{C_0}{k} \right).$$

With the goal of summing up these expressions from $1$ to infinity, we claim that there is some constant $M > 0$ such that for any $C' \in \mathbb{R}$, and $2|C'| \leq n < N$ we have that

$$\left| \sum_n^N \ln \left( 1 + \frac{C'}{k} \right) - C' \ln \left( \frac{N}{n-1} \right) \right| \leq M. \tag{13}$$

Given this claim we conclude that

$$- \left( C_0 \ln \left( N \right) + \left[ M - C_0 \ln \left( 2C_0 \right) \right] \right) \leq \sum_{k=2C_0+1}^{N} \ln \left| \frac{b \left( k \right)}{Bk^d} \right| \leq C_0 \ln \left( N \right) + \left[ M - C_0 \ln \left( 2C_0 \right) \right].$$

For another $C$ large enough (independent of $N$), we can start the summation from $k = 1$ to get

$$-C \left( \ln \left( N \right) + 1 \right) \leq \sum_{k=1}^{N} \ln \left| \tilde{b} \left( k \right) \right| \leq C \left( \ln \left( N \right) + 1 \right).$$

Finally, exponenting it back we get the result we wanted:

$$\left( Ne \right)^{-C} \leq \prod_{k=1}^{N} \left| \tilde{b} \left( k \right) \right| \leq \left( Ne \right)^{C}.$$

We are left to prove Equation (13).

Using the Taylor expansion of $\ln \left( 1 + x \right)$ for $|x| \leq \frac{1}{2}$, we know that there is some large enough $0 < M_0$ such that

$$\left| \ln \left( 1 + x \right) - x \right| \leq M_0 x^2.$$

It follows that for $2 \left| C' \right| \leq n < N$ we have

$$\left| \sum_{k=n}^{N} \left( \ln \left( 1 + \frac{C'}{k} \right) - \frac{C'}{k} \right) \right| \leq M_0 C'^2 \sum_{n}^{N} \frac{1}{k^2} \leq M_0 C'^2 \zeta \left( 2 \right).$$

In addition, we have that $\left| \sum_n^N \frac{1}{k} - \int_{n-1}^N \frac{1}{x} \mathrm{d}x \right| \leq 1$, and

$$\int_{n-1}^{N} \frac{1}{x} \mathrm{d}x = \ln \left( \frac{N}{n-1} \right).$$

Therefore

$$\left| \sum_{n}^{N} \ln \left( 1 + \frac{C'}{k} \right) - C' \ln \left( \frac{N}{n-1} \right) \right| \leq \left| C' \right| + M_0 C'^2 \zeta \left( 2 \right)$$

is uniformly bounded.

### F.3   Growth rate of $q_n$

The sequence $q_n$ satisfies the linear recurrence

$$q_{n+1} = a \left( n \right) q_n + b \left( n \right) q_{n-1},$$

or in matrix form

$$\left( q_n, q_{n+1} \right) = \left( q_{n-1}, q_n \right) \overbrace{\begin{pmatrix} 0 & b \left( n \right) \\ 1 & a \left( n \right) \end{pmatrix}}^{M(n)}.$$

If both $a \left( x \right), b \left( x \right)$ are constant, and therefore $M = M \left( n \right)$ is a constant matrix, then this problem reduces to simply $\left( q_n, q_{n+1} \right) = \left( q_0, q_1 \right) M^n$. Its a standard exercise to approximate $q_n$ using the eigenvectors decomposition of $M$. However, in general not only $M \left( n \right)$ is non-constant, its entries have different orders of magnitude.

Thus, we would like to move to an "equivalent" system where at the very least $M \left( n \right)$ converges to some matrix $M_\infty$, and then hope to show that the behavior of $q_n$ can be read from the system with $M_\infty^n$. This equivalent system will be built in two steps: first we "balance" the matrix, so its coordinates growth rate are the same, and then taking it outside as a scalar, the remaining sequence of matrices will converge.

### F.3.1 Matrix balancing

This balancing is split into two cases according to the degrees of $d_a = \deg(a(x))$, $d_b = \deg(b(x))$.

Let $d = \max\left\{d_a, \frac{1}{2}d_b\right\}$ and denote by $A, B$ the coefficients of $x^d, x^{2d}$ of $a(x), b(x)$ respectively. Note that both $A, B$ are either the corresponding leading coefficients or zero, depending on whether $d_a = d$, respectively $d_b = 2d$. If $2d_a < d_b$ and $d_b$ is odd, then $d_a < d = \frac{d_b}{2}$, and we still consider $A$ to be zero. Regardless of the choice of $d$, we see that at least one of $A$ or $B$ is not zero (and both if $2d_a = d_b$, which we call a "balanced" PCF).

With this choice, taking $\tilde{q}_n = \frac{q_n}{(n!)^d}$, we obtain the linear recurrence

$$\tilde{q}_{n+1} = \frac{a(n)}{(n+1)^d}\tilde{q}_n + \frac{b(n)}{(n(n+1))^d}\tilde{q}_{n-1}.$$

Letting $\tilde{a}(n) = \frac{a(n)}{(n+1)^d}$ and $\tilde{b}(n) = \frac{b(n)}{(n(n+1))^d}$, by our choice of $d$ we see that the coefficient or the recurrence converge, and not both to zero:

$$\lim_{n\to\infty} \tilde{a}(n) = A$$
$$\lim_{n\to\infty} \tilde{b}(n) = B.$$

Here too we can also write it in a matrix form, namely

$$(\tilde{q}_n, \tilde{q}_{n+1}) = (\tilde{q}_{n-1}, \tilde{q}_n)\begin{pmatrix} 0 & \tilde{b}(n) \\ 1 & \tilde{a}(n) \end{pmatrix}.$$

We now have a limit matrix, and the dynamics of such a matrix is well known. If both eigenvalues are real which are distinct in absolute value, then we expect exponential convergence. If both are non real, and therefore complex conjugate we expect it to behave like a rotation, and therefore will not converge. In both of these cases, since the eigenvalues are distinct in the limit, this holds for almost all $n$, so this behavior should hold in general.

In the discriminant zero, the situation is much more delicate, since we can converge to zero in many ways. For example, the discriminant along the way can be negative, positive or zero. In this notes we will restrict the study only to the two real eigenvalues with different absolute values.

### F.3.2 Asymptotics of the continued fraction recurrence

The main goal of this section is to approximate the growth rate of a solution $u_n$ to the recurrence

$$u_{n+1} = a_n u_n + b_n u_{n-1},$$

where both $a_n, b_n$ converge (and not both to zero) or in matrix form

$$\begin{pmatrix} v_n & v_{n+1} \end{pmatrix} = \begin{pmatrix} v_{n-1} & v_n \end{pmatrix} M_n \quad , \quad M_n = \begin{pmatrix} 0 & b_n \\ 1 & a_n \end{pmatrix},$$

where $M_n \to M := \begin{pmatrix} 0 & b \\ 1 & a \end{pmatrix}$.

The first step is the standard conjugation to a simpler matrix. Indeed, if $D = PMP^{-1}$ is simpler, e.g. diagonal, then $D_n := PM_nP^{-1} \to D$, and $\prod_1^n M_i = P^{-1}\prod_1^n D_i P$, so we more or less need to understand $\prod_1^n D_i$.

In the constant diagonal case $D_n = \begin{pmatrix} \lambda_1 & 0 \\ 0 & \lambda_2 \end{pmatrix}$ with $\lambda_1 > |\lambda_2|$, we expect that for almost every initial condition $\left\|(\alpha_1, \beta_1)D^k\right\| \sim \lambda_1^k$. This is true as long as the initial vector is not in $\mathbb{R}\cdot e_2$, and we have similar behaviour for other type of matrices. When the $D_n$ are not constant, we need to take a little bit more care. The image you should have in mind is the following:

Instead of the two eigenvectors being on the $X$ and $Y$ axes, they only converge to it, so we only know that they are somewhere inside the red and blue regions. Thus, to understand this system we first need a **separation condition** saying that these regions are disjoint. Assuming the $X$-axis is the

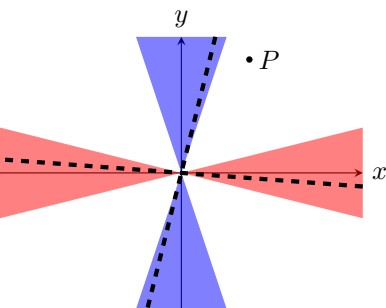

Figure 8: Convergence with variable coefficients

pulling axis (larger eigenvalue), we will need at least one point outside the error region around the $Y$ axis, which we call the **initial condition**. Once both these conditions hold, a standard investigation of diagonalizable product will show that the point's orbit converge towards the eigenvector in the $X$-region. As this region shrinks to $X$ in the limit, we see that the limit of the orbit is there as well.

**Lemma 1.** Suppose that $D_n \to D$ where $D = \begin{pmatrix} \lambda_+ & 0 \\ 0 & \lambda_- \end{pmatrix}$ with $0 \leq \left| \frac{\lambda_-}{\lambda_+} \right| < 1$ and let $\kappa_n = \frac{1}{|\lambda_+|} \max_{k \geq n} \|D_k - D\|_\infty$.

Fix some initial $z_1$ and let $z_k = (z_1) \prod_1^{k-1} D_n$. Assuming that for some $n$ we have

- **Separation condition**: $\left| \frac{\lambda_-}{\lambda_+} \right| + 4\kappa_n < 1$ and

- **Initial condition**: $|z_n| < \frac{\mu_n + \sqrt{(\mu_n - 2\kappa_n)(\mu_n + 2\kappa_n)}}{2\kappa_n}$, $\mu_n = 1 - \left| \frac{\lambda_-}{\lambda_+} \right| - 2\kappa_n$.

Then $\lim_{k \to \infty} |z_k| = 0$.

**Remark 2.** Note that $\frac{\mu_n + \sqrt{(\mu_n - 2\kappa_n)(\mu_n + 2\kappa_n)}}{2\kappa_n} \sim \frac{1 - |\lambda|}{\kappa_n} \to \infty$ as $\kappa_n \to 0$, so this initial condition becomes easier to satisfy as $n \to \infty$.

*Proof.* First, proving the claim for $\frac{1}{\lambda_+} D_n$ instead of $D_n$, we may assume that the limit is $D = \begin{pmatrix} 1 & 0 \\ 0 & \lambda \end{pmatrix}$ where $\lambda = \frac{\lambda_-}{\lambda_+}$.

Next, note that whenever $\mu := 1 - |\lambda| - 2\kappa > 2\kappa > 0$, we have that $\sqrt{(\mu - 2\kappa)(\mu + 2\kappa)} = \sqrt{\mu^2 - 4\kappa^2} < \mu$. Setting

$$\nu_\pm(\kappa) = \frac{\mu_\kappa \pm \sqrt{(\mu_\kappa - 2\kappa)(\mu_\kappa + 2\kappa)}}{2\kappa},$$

we get that $0 < \nu_-(\kappa) < \nu_+(\kappa)$ are real numbers, and the condition in the assumption is $|z_n| < \nu_+(\kappa_n)$. Our main goal is to prove our process satisfies:

1. $|z_k| < \nu_+(\kappa_n)$ for all $k \geq n$ and,

2. We have $\limsup_k |z_k| \leq \nu_-(\kappa_n)$.

Assuming these two steps are true, the full proof is not too far behind. Indeed, since $D_n \to D$, the sequence $\kappa_n := \sup_{k \geq n} \|D_n - D\|$ converges to zero, and note that as $\kappa_n \to 0$ we get that $\nu_+(\kappa_n) \nearrow \infty$ and $\nu_-(\kappa_n) \searrow 0$. Assuming step (1), for $k \geq m \geq n$ we have $|z_k| < \nu_+(\kappa_n) \leq \nu_+(\kappa_m)$, and by step (2) we get that $\limsup_k |z_k| \leq \nu_-(\kappa_m) \to 0$.

For the remaining of the proof, without loss of generality we may assume that $n = 1$ and just write $\kappa, \mu$ instead of $\kappa_n, \mu_n$.

To prove these two steps, consider the change from $z_k$ to $z_{k+1}$. Writing $D_k = \begin{pmatrix} 1+\varepsilon_{1,1} & \varepsilon_{1,2} \\ \varepsilon_{2,1} & \lambda+\varepsilon_{2,2} \end{pmatrix}$, since $z_{k+1} = (z_k) D_k$ and $\|D - D_k\|_\infty \le \kappa$, we get that

$$|z_{k+1}| = \left| \frac{\varepsilon_{1,2} + z_k (\lambda + \varepsilon_{2,2})}{(1 + \varepsilon_{1,1}) + z_k \varepsilon_{2,1}} \right| \le \frac{\kappa + |z_k| (|\lambda| + \kappa)}{1 - \kappa - \kappa |z_k|}.$$

Note that the final denominator is positive, so that the last inequality is valid. Indeed, using the conditions of the claim we get

$$1 - \kappa (1 + |z_k|) \ge 1 - \kappa \left( 1 + \frac{\mu + \sqrt{(\mu - 2\kappa)(\mu + 2\kappa)}}{2\kappa} \right) > 1 - (\kappa + \mu) = |\lambda| + \kappa > 0.$$

Thus, we can rewrite the inequality as

$$|z_{k+1}| \le M_\varepsilon (|z_k|), \quad M_\varepsilon = \begin{pmatrix} |\lambda| + \kappa & \kappa \\ -\kappa & 1 - \kappa \end{pmatrix}. \tag{14}$$

The goal now is to show that if $|z_k|$ is "large", then $|z_{k+1}|$ is much smaller, and if $|z_k|$ is small, then $|z_{k+1}|$ cannot increase too much.

A simple computations shows that the eigenvalues of this matrix are

$$\gamma_\pm = \frac{|\lambda| + 1 \pm \sqrt{(\mu + 2\kappa)(\mu - 2\kappa)}}{2},$$

and since $\sqrt{(\mu + 2\kappa)(\mu - 2\kappa)} \le \mu \le 1 - |\lambda|$, we get that

$$\gamma_+ > \gamma_- > 0.$$

Finally, the corresponding (right) eigenvectors are

$$v_\pm = \begin{pmatrix} \nu_\mp \\ 1 \end{pmatrix}.$$

To simplify the notations, let us conjugate by the matrix $T = \begin{pmatrix} \nu_+ & \nu_- \\ 1 & 1 \end{pmatrix}$ to obtain

$$T^{-1} M_\varepsilon T = \begin{pmatrix} \gamma_- & 0 \\ 0 & \gamma_+ \end{pmatrix}.$$

Note that the Mobius map

$$T^{-1}(z) := \frac{1}{\nu_+ - \nu_-} \begin{pmatrix} 1 & -\nu_- \\ -1 & \nu_+ \end{pmatrix}(z) = -\frac{z - \nu_-}{z - \nu_+} = -1 + \frac{\nu_- - \nu_+}{z - \nu_+}$$

sends $\nu_- \mapsto 0$, $\nu_+ \mapsto \infty$ and $0 \mapsto -\frac{\nu_-}{\nu_+} < 0$. In particular, it is monotone increasing on $[0, \nu_+)$, so that our two steps from above are equivalent to

1. $T^{-1}(|z_k|) \in [-\frac{\nu_-}{\nu_+}, \infty)$,

2. $\limsup_k T^{-1}(|z_k|) \in \left[ -\frac{\nu_-}{\nu_+}, 0 \right]$,

and the claim's original assumption is that $T^{-1}(|z_1|) \in [-\frac{\nu_-}{\nu_+}, \infty)$. However, now this claim is simple, since in these notations we get that

$$T^{-1}(M_\varepsilon (|z_k|)) = (T^{-1} M_\varepsilon T)(T^{-1}(|z_k|)) = \frac{\gamma_-}{\gamma_+} \cdot T^{-1}(|z_k|),$$

and $0 < \frac{\gamma_-}{\gamma_+} < 1$. Thus, if $T^{-1}(|z_k|) \in [-\frac{\nu_-}{\nu_+}, \infty)$, then so is $T^{-1}(M_\varepsilon (|z_k|)) \in [-\frac{\nu_-}{\nu_+}, \infty)$, so by Equation (14) and the monotonicity of $T$, we obtain that

$$T^{-1}(|z_{k+1}|) \le \frac{\gamma_-}{\gamma_+} \cdot T^{-1}(|z_k|),$$

which implies the two steps. $\qquad \square$

Returning back to the recursion, we get the following.

**Theorem 1.** Suppose that we have a solution to the recurrence $v_{n+1} = a_n v_n + b_n v_{n-1}$, where $a_n \to a, b_n \to b$ and suppose that $\lambda_\pm$ are the roots of $x^2 = ax + b$ with $0 \le |\lambda_-| < \lambda_+$. Writing $\kappa'_n = \frac{1}{|\lambda_+|} \max_{k \ge n} \max \{|a_k - a|, |b_k - b|\}$ and $C(\lambda_\pm) := \frac{1+|\lambda_+|}{|\lambda_+ - \lambda_-|}$, Assume that for some $n$ we have

- **Separation condition**: $\left|\frac{\lambda_-}{\lambda_+}\right| + 4C(\lambda_\pm) \kappa'_n < 1$ and

- **Initial condition**: $\left|\lambda_- - \frac{v_n}{v_{n-1}}\right| \ge C(\lambda_\pm) \kappa'_n \frac{|\lambda_+ - \lambda_-|}{1 - \left|\frac{\lambda_-}{\lambda_+}\right| - 4C(\lambda_\pm)\kappa'_n}$,

Then
$$\frac{v_n}{v_{n-1}} \to \lambda_+.$$

**Proof 2.** Set $M_n = \left(\begin{smallmatrix} 0 & b_n \\ 1 & a_n \end{smallmatrix}\right)$ and $M = \left(\begin{smallmatrix} 0 & b \\ 1 & a \end{smallmatrix}\right)$ as in the beginning of this section. With $P = \left(\begin{smallmatrix} 1 & \lambda_+ \\ 1 & \lambda_- \end{smallmatrix}\right)$ and $P^{-1} = \frac{1}{\lambda_- - \lambda_+} \left(\begin{smallmatrix} \lambda_- & -\lambda_+ \\ -1 & 1 \end{smallmatrix}\right)$ we have that $D = PMP^{-1} = \left(\begin{smallmatrix} \lambda_+ & 0 \\ 0 & \lambda_- \end{smallmatrix}\right)$. We would like to apply Lemma 1 to the matrices $D_n = PM_nP^{-1}$.

For the **separation condition** on the infinity norm, we have

$$\left\|PM_nP^{-1} - D\right\|_\infty = \left\|P(M_n - M)P^{-1}\right\|_\infty = \frac{1}{|\lambda_- - \lambda_+|} \left\|\left(\begin{smallmatrix} 1 & \lambda_+ \\ 1 & \lambda_- \end{smallmatrix}\right) \left(\begin{smallmatrix} 0 & b_n - b \\ 0 & a_n - a \end{smallmatrix}\right) \left(\begin{smallmatrix} \lambda_- & -\lambda_+ \\ -1 & 1 \end{smallmatrix}\right)\right\|_\infty$$

$$= \frac{1}{|\lambda_- - \lambda_+|} \left\|\left(\begin{smallmatrix} 1 & \lambda_+ \\ 1 & \lambda_- \end{smallmatrix}\right) \left(\begin{smallmatrix} b - b_n & b_n - b \\ a - a_n & a_n - a \end{smallmatrix}\right)\right\|_\infty \le \overbrace{\frac{1 + |\lambda_+|}{|\lambda_+ - \lambda_-|}}^{C(\lambda_\pm)} \|M_n - M\|_\infty.$$

Thus, the separation condition of this theorem implies the separation condition of Lemma 1:

$$\left|\frac{\lambda_-}{\lambda_+}\right| + 4\kappa_n \le \left|\frac{\lambda_-}{\lambda_+}\right| + 4C(\lambda_\pm) \frac{1}{|\lambda_+|} \max_{k \ge n} \|M_n - M\|_\infty < 1$$

Next, for the **initial condition**, setting

$$(v_{k-1}\ v_k) := (v_0\ v_1) \left(\prod_1^{k-1} M_n\right) = (v_0\ v_1) P^{-1} \left(\prod_1^{k-1} D_n\right) P$$

we have

$$(\alpha_k, \beta_k) = (v_0\ v_1) P^{-1} \left(\prod_1^{k-1} D_n\right) = (v_{k-1}, v_k) P^{-1}.$$

Setting $z_n = \frac{\beta_n}{\alpha_n}$, we get that

$$|z_n| = \left|\frac{\beta_n}{\alpha_n}\right| = \left|\frac{-\lambda_+ v_{n-1} + v_n}{\lambda_- v_{n-1} - v_n}\right| = \left|1 + \frac{\lambda_+ - \lambda_-}{\lambda_- - \frac{v_n}{v_{n-1}}}\right| \le 1 + \left|\frac{\lambda_+ - \lambda_-}{\lambda_- - \frac{v_n}{v_{n-1}}}\right| = (*).$$

Using the assumption that $\left|\lambda_- - \frac{v_n}{v_{n-1}}\right| \ge C(\lambda_\pm) \kappa'_n \frac{|\lambda_+ - \lambda_-|}{1 - \left|\frac{\lambda_-}{\lambda_+}\right| - 4C(\lambda_\pm)\kappa'_n} \ge \kappa_n \frac{|\lambda_+ - \lambda_-|}{1 - \left|\frac{\lambda_-}{\lambda_+}\right| - 4\kappa_n}$, we see that the expression above is

$$(*) \le 1 + \frac{|\lambda_+ - \lambda_-|}{\kappa_n \frac{|\lambda_+ - \lambda_-|}{1 - \left|\frac{\lambda_-}{\lambda_+}\right| - 4\kappa_n}} = 1 + \frac{1 - \left|\frac{\lambda_-}{\lambda_+}\right| - 4\kappa_n}{\kappa_n} = \frac{2\mu_n - 2\kappa_n}{2\kappa_n} < \frac{\mu_n + \sqrt{(\mu_n - 2\kappa_n)(\mu_n + 2\kappa_n)}}{2\kappa_n}.$$

This was the second condition needed for Lemma 1, so we can now conclude that

$$\left|1 + \frac{\lambda_+ - \lambda_-}{\lambda_- - \frac{v_n}{v_{n-1}}}\right| = \left|\frac{\beta_n}{\alpha_n}\right| \to 0$$

which implies that $\frac{v_n}{v_{n-1}} \to \lambda_+$.

## F.4 Conclusion

We return now to the original problem with $\alpha = \mathbb{K}_1^\infty \frac{b(n)}{a(n)}$ and assume that $a(x), b(x)$ have degrees $d_a, d_b$. As mentioned before, we split our study into two cases:

**The balanced case**

Assume that $d_b = 2d_a = 2d$, and let $A, B$ be the leading coefficients of $a(x), b(x)$ respectively.

In this case the limit matrix is $M_\infty = \begin{pmatrix} 0 & B \\ 1 & A \end{pmatrix}$, and we assume that the roots $\lambda_\pm$ of $x^2 = Ax + B$ satisfy $0 < |\lambda_-| < \lambda_+$. Using Theorem 1 once the two conditions hold, we obtain

$$\frac{q_{n+1}}{q_n}(n+1)^d = \frac{q_{n+1}/(n+1)!^d}{q_n/n!^d} \to \lambda_+,$$

implying that $q_n = n!^d \lambda_+^n \exp(o(n))$. As for the product of the $b(k)$, using Claim 1 we have that

$$\prod_{k=1}^{N} |b(k)| = \exp(o(N)) \cdot B^N \cdot N!^{2d}.$$

Putting them together as in the error rate expression, we get :

$$\frac{\prod_{k=1}^{m-1}|b(k)|}{|q_{m-1}q_m|} = \frac{|B|^{m-1} \cdot (m-1)!^{2d}}{(m-1)!^d(m)!^d\lambda_+^{2m-1}} \exp(o(m)) = (*).$$

Note that $|B| = |\det(M_\infty)| = |\lambda_-\lambda_+|$, so that the expression above is

$$(*) = |\lambda_-/\lambda_+|^m \cdot \exp(o(m)) = \exp(m\log|\lambda_-/\lambda_+| + o(m)).$$

Thus, for given $\varepsilon > 0$ where $\left|\frac{\lambda_-}{\lambda_+}\right| + \varepsilon < 1$, and for any $m$ large enough we see that $(*) \leq \left(\left|\frac{\lambda_-}{\lambda_+}\right| + \varepsilon\right)^{m-1}$. We conclude that the error rate for all $n$ large enough is bounded from above by

$$\left|\frac{p_n}{q_n} - \alpha\right| \leq \sum_{m=n+1}^{\infty} \frac{\prod_{k=1}^{m-1}|b(k)|}{|q_{m-1}q_m|} \leq \sum_{m=n+1}^{\infty} \left(\left|\frac{\lambda_-}{\lambda_+}\right| + \varepsilon\right)^{m-1} = \left(\left|\frac{\lambda_-}{\lambda_+}\right| + \varepsilon\right)^n \frac{1}{1 - \left(\left|\frac{\lambda_-}{\lambda_+}\right| + \varepsilon\right)}.$$

It follows that

$$\ln\left|\frac{p_n}{q_n} - \alpha\right| \leq n\ln\left(\left(\left|\frac{\lambda_-}{\lambda_+}\right| + \varepsilon\right)\right) - \ln\left(1 - \left(\left|\frac{\lambda_-}{\lambda_+}\right| + \varepsilon\right)\right) \sim n\ln\left(\left|\frac{\lambda_-}{\lambda_+}\right|\right).$$

**The unbalanced case**

Suppose now that $d_b < 2d_a = 2d$, so that $B = \lim_{n\to\infty} \frac{b(n)}{(n(n+1))^{d_a}} = 0$. This time the two roots of $x^2 = Ax + 0$ are $\lambda = 0, A$. If needed, we can use a simple continued fraction inflation $\mathbb{K}_1^\infty \frac{(-1)^2 b(n)}{(-1)a(n)}$ and assume that $A > 0$. Using Theorem 1, if the two conditions hold, we obtain $q_n = n!^d A^n \exp(o(n))$.

Letting $\hat{B}$ be the leading coefficient of $b(x)$ in absolute value, Claim 1 implies that

$$\prod_{k=1}^{N} |b(k)| = \exp(o(N)) \cdot \hat{B}^N \cdot N!^{d_b}.$$

Again, together we obtain that

$$\frac{\prod_{k=1}^{m-1}|b(k)|}{|q_{m-1}q_m|} = \frac{\hat{B}^{m-1} \cdot (m-1)!^{d_b}}{(m-1)!^d m!^d A^{2m-1}} \exp(o(m)) = \frac{1}{(m-1)!^{2d_a-d_b}} \cdot \left(\frac{\hat{B}}{A^2}\right)^m \exp(o(m))$$

Similarly to the previous case, given $\varepsilon > 0$, and using the fact that $2d_a - d_b \geq 1$, for all $n$ large enough we obtain

$$\left|\frac{p_n}{q_n} - \alpha\right| \leq \sum_{m=n+1}^{\infty} \frac{\prod_{k=1}^{m-1}|b(k)|}{|q_{m-1}q_m|} \leq \sum_{m=n+1}^{\infty} \frac{1}{(m-1)!^{2d_a-d_b}} \cdot \left(\frac{\hat{B}}{A^2} + \varepsilon\right)^{m-1}$$

$$= \frac{1}{n!^{2d_a-d_b}} \left(\frac{\hat{B}}{A^2} + \varepsilon\right)^n \sum_{m=0}^{\infty} \left(\frac{n!}{(n+m)!}\right)^{2d_a-d_b} \cdot \left(\frac{\hat{B}}{A^2} + \varepsilon\right)^m$$

$$\leq \frac{1}{n!^{2d_a-d_b}} \left(\frac{\hat{B}}{A^2} + \varepsilon\right)^n \left[\sum_{m=0}^{\infty} \left(\frac{1}{m!}\right)^{2d_a-d_b} \cdot \left(\frac{\hat{B}}{A^2} + \varepsilon\right)^m\right].$$

The infinite sum in the last expression converges to some finite limit $\tilde{C}$, so we conclude that

$$\ln\left|\frac{p_n}{q_n} - \alpha\right| \leq (d_b - 2d_a)\ln(n!) + n\ln\left|\frac{\hat{B}}{A^2} + \varepsilon\right| + \ln\left|\tilde{C}\right| \sim (d_b - 2d_a)\,n \cdot \ln|n|.$$

# G   List of Proven Formulas

Table 5: This is a table showing the automatically generated conjectured formulas that were analytically proven. For the method of proving, see appendix A.3. Note that lines **1,4** and **7** are proven infinite families of formulas, generalized from the cases found in the dataset.

| $A_n$ | $B_n$ | $h_1(n)$ | $h_2(n)$ | $f(n)$ | Known limit of resulting infinite sum |
|---|---|---|---|---|---|
| $\omega + 1$ | $n^2 + \omega n$ | $-n$ | $n + \omega$ | $1$ | $(\omega+1)/{}_2F_1(1,1,\omega+2,-1)$ |
| $5$ | $n^2 + 4n$ | $-n$ | $n + 4$ | $1$ | $-\frac{12}{131+192\log(2)}$ |
| $4$ | $n^2 + 3n$ | $-n$ | $n + 3$ | $1$ | $\frac{3}{24\log(2)-16}$ |
| $\omega$ | $(\omega n + 1)^2$ | $-\omega n - 1$ | $\omega n + 1$ | $1$ | $(\omega+1)/{}_2F_1(1,\frac{\omega+1}{\omega},\frac{2\omega+1}{\omega},-1)-1$ |
| $2$ | $4n^2 + 4n + 1$ | $-2n - 1$ | $2n + 1$ | $1$ | $\frac{\pi}{4-\pi}$ |
| $1$ | $n^2 + 2n + 1$ | $-n - 1$ | $n + 1$ | $1$ | $\frac{\log(2)}{1-\log(2)}$ |
| $\omega$ | $n^2 + (\omega+1)n + \omega$ | $-n - 1$ | $n + \omega$ | $1$ | $(\omega+1)/{}_2F_1(1,2,\omega+2,-1)-1$ |
| $3$ | $n^2 + 4n + 3$ | $-n - 1$ | $n + 3$ | $1$ | $\frac{4}{34-48\log(2)}-1$ |
| $2$ | $n^2 + 3n + 2$ | $-n - 1$ | $n + 2$ | $1$ | $\frac{3}{9-12\log(2)}-1$ |
| $5$ | $n^2 + 2n$ | $n$ | $n + 2$ | $n + \frac{3}{2}$ | $\frac{2}{17-24\log(2)}$ |
| $5$ | $n^2 + 4n + 3$ | $-n - 1$ | $n + 3$ | $n + \frac{5}{2}$ | $\frac{3(17-24\log(2))}{120\log(2)-83}$ |
| $4$ | $n^2 + n$ | $-n$ | $n + 1$ | $n + 1$ | $\frac{1}{3-4\log(2)}$ |
| $4$ | $n^2 + 3n + 2$ | $-n - 1$ | $n + 2$ | $n + 2$ | $\frac{6-8\log(2)}{16\log(2)-11}$ |
| $4$ | $4n^2 + 4n$ | $-2n$ | $2n + 2$ | $1$ | $\frac{2}{2\log(2)-1}$ |
| $3$ | $n^2$ | $-n$ | $n$ | $n + \frac{1}{2}$ | $\frac{1}{1-\log(2)}$ |
| $3$ | $n^2 + 2n + 1$ | $-n - 1$ | $n + 1$ | $n + \frac{3}{2}$ | $\frac{1-\log(2)}{3\log(2)-2}$ |
| $3$ | $n^2 + 4n + 4$ | $-n - 2$ | $-n + 2$ | $n + \frac{5}{2}$ | $\frac{4(3\log(2)-2)}{7-10\log(2)}$ |
| $1$ | $n^2 + 4n + 4$ | $n + 2$ | $-n - 2$ | $1$ | $\frac{2}{2\log(2)-1}-2$ |
| $4$ | $4n^2 - 1$ | $-2n + 1$ | $2n + 1$ | $1$ | $\frac{\pi+2}{\pi-2}$ |
| $2$ | $4n^2 - 4n - 1$ | $-2n + 1$ | $2n - 1$ | $1$ | $\frac{4}{\pi}+1$ |

| | | | | | |
|---|---|---|---|---|---|
| $5$ | $4n^2 + 2n - 2$ | $-2n - 1$ | $2n + 2$ | $1$ | $\frac{22 + 12\sqrt{2}}{7}$ |
| $5$ | $4n^2 + 2n$ | $-2n - 1$ | $2n$ | $n + \frac{3}{4}$ | $3 + 2\sqrt{2}$ |
| $3$ | $4n^2 - 2n$ | $-2n + 1$ | $2n$ | $1$ | $2 + \sqrt{2}$ |
| $2n^2 + 2n + 1$ | $-n^4$ | $n^2$ | $n^2$ | $1$ | $\frac{1}{\zeta(2)}$ |
| $2n + 1$ | $n^4$ | $-n^2$ | $n^2$ | $1$ | $\frac{2}{\zeta(2)}$ |
| $n^2 + n + 1$ | $n^4 + n^2 + 1$ | $\phi(n^2 + n + 1)$ | $(\phi - 1)(-n^2 + n - 1)$ | $1$ | $\phi$ |
| $2n^2 + 2n + 2$ | $n^4 + n^2 + 1$ | $(\sqrt{2} + 1)(n^2 + n + 1)$ | $(\sqrt{2} - 1)(-n^2 + n - 1)$ | $1$ | $1 + \sqrt{2}$ |
| $2n^2 + 2n + 2$ | $2n^4 + 2n^2 + 2$ | $(\sqrt{3} + 1)(n^2 + n + 1)$ | $(\sqrt{3} - 1)(-n^2 + n - 1)$ | $1$ | $1 + \sqrt{3}$ |
| $-5$ | $n^2 + 2n$ | $-n$ | $-n - 2$ | $n + \frac{3}{2})$ | $-\frac{5}{\frac{85}{2} - 60\log(2)}$ |
| $-5$ | $n^2 + 4n$ | $n$ | $-n - 4$ | $1$ | $-\frac{5}{-\frac{655}{12} + 80\log(2)}$ |
| $-4$ | $n^2 + n$ | $n$ | $-n - 1$ | $n + 1$ | $-\frac{4}{12 - 16\log(2)}$ |
| $-4$ | $n^2 + 3n$ | $n$ | $-n - 3$ | $1$ | $-\frac{4}{-\frac{64}{3} + 32\log(2)}$ |
| $-4$ | $n^2 + 3n + 2$ | $n + 1$ | $-n - 2$ | $n + 2$ | $\frac{1}{2} - \frac{9}{2(-99 + 144\log(2))}$ |
| $-4$ | $n^2 + 5n + 4$ | $n + 1$ | $-n - 4$ | $1$ | $1 - \frac{5}{\frac{335}{3} - 160\log(2)}$ |
| $-4$ | $4n^2 + 4n$ | $2n$ | $-2n - 2$ | $1$ | $-\frac{4}{-2 + 4\log(2)}$ |
| $-3$ | $n^2$ | $n$ | $-n - 2$ | $1$ | $-\frac{3}{3 - 3\log(2)}$ |
| $-3$ | $n^2 + 2n$ | $n$ | $-n - 2$ | $1$ | $-\frac{3}{-\frac{15}{2} + 12\log(2)}$ |
| $-3$ | $n^2 + 2n + 1$ | $n + 1$ | $-n - 1$ | $n + \frac{3}{2}$ | $\frac{1}{3} - \frac{10}{3(-20 + 30\log(2))}$ |
| $-3$ | $n^2 + 4n + 3$ | $n + 1$ | $-n - 3$ | $1$ | $1 - \frac{4}{34 - 48\log(2)}$ |
| $-3$ | $n^2 + 4n + 4$ | $n + 2$ | $n - 2$ | $n + \frac{5}{2}$ | $\frac{6}{5} - \frac{21}{5\left(\frac{147}{2} - 105\log(2)\right)}$ |
| $-2$ | $n^2 + n$ | $n$ | $-n - 1$ | $1$ | $-\frac{2}{-2 + 4\log(2)}$ |
| $-2$ | $n^2 + 3n + 2$ | $n + 1$ | $-n - 2$ | $1$ | $1 - \frac{3}{9 - 12\log(2)}$ |
| $-2$ | $4n^2$ | $2n$ | $-2n$ | $1$ | $-\frac{1}{\log(2)}$ |
| $-1$ | $n^2$ | $n$ | $-n$ | $1$ | $-\frac{1}{\log(2)}$ |
| $-1$ | $n^2 + 2n + 1$ | $n + 1$ | $-n - 1$ | $1$ | $1 - \frac{2}{2 - 2\log(2)}$ |
| $-1$ | $n^2 + 4n + 4$ | $-n - 2$ | $n + 2$ | $1$ | $2 - \frac{3}{-\frac{3}{2} + 3\log(2)}$ |
| $-4$ | $4n^2 - 1$ | $2n - 1$ | $-2n - 1$ | $1$ | $-\frac{3}{-\frac{3}{2} + \frac{3\pi}{4}} - 1$ |
| $-2$ | $4n^2 - 4n + 1$ | $2n - 1$ | $-2n + 1$ | $1$ | $-\frac{4}{\pi} - 1$ |
| $-2$ | $4n^2 + 4n + 1$ | $2n + 1$ | $-2n - 1$ | $1$ | $1 - \frac{3}{3 - \frac{3\pi}{4}}$ |
| $-5$ | $4n^2 + 2n - 2$ | $2n + 1$ | $-2n - 2$ | $1$ | $-\frac{4}{-\frac{20}{3} + \frac{16\sqrt{2}}{3}} - 1$ |
| $-5$ | $4n^2 + 2n$ | $2n + 1$ | $-2n$ | $n + \frac{3}{4}$ | $-\frac{32\Gamma\left(\frac{11}{4}\right)}{9\left(-\frac{-56\Gamma\left(\frac{5}{4}\right)\Gamma\left(\frac{7}{4}\right)}{\pi} - 14\right)\Gamma\left(\frac{3}{4}\right)} - \frac{1}{3}$ |
| $-3$ | $4n^2 - 2n$ | $2n - 1$ | $-2n$ | $1$ | $-\frac{2}{-2 + 2\sqrt{2}} - 1$ |

