# OpenReview forum: "Unsupervised Discovery of Formulas for Mathematical Constants"
_NeurIPS.cc/2024/Conference — NeurIPS 2024 poster_

### Official Review · Reviewer_Dgb8 · 2024-07-01

**Soundness:** 3
**Presentation:** 3
**Contribution:** 2
**Rating:** 5
**Confidence:** 3

**Summary:**

The authors propose an algorithm to filter, cluster and label polynomial continued fractions. It is based on several features linked to the asymptotic behaviour of their rational approximations. The authors detail how these are computed. They apply this algorithm to a large set of formulae ; they discuss the properties of the obtained clusters and highlight some novel PCFs linked to known mathematical constants.

**Strengths:**

This work seems to be an interesting application of unsupervised data analysis to fundamental science. It proposes a way to build relevant metrics from sequences of numbers, and to numerically estimate them.

More generally, the structures revealed on fig. 3 and 4 are intriguing and the article is well written.

**Weaknesses:**

The claim that "we connect the challenge of formula creation to modern approaches in AI for Science" seems a bit bold. To my understanding this work does not involve modern AI in the sense that it requires manual and careful feature extraction and that nothing is learnt.

**Questions:**

- Part 3.4 was a bit confusing. The Blind-δ Algorithm is presented as a proxy for delta eq. 4. Is it not rather to estimate the approximation error epsilon ?

- There is an inconsistency : part 3.2 it is n^beta vs table 1 P(n) and no beta ; is it because beta is hard to estimate ? is it usefull for clustering ?

- Could the authors comment on the fact that the predicted delta eq. 5 can be quite far from the actual delta (fig. 2a, fig. 2b) ? Yet it seems it provides relevant information for clustering ; what does it capture ?

- More generally, did the authors try to extract more features, other metrics, not necessary having a mathematical interpretation, in an automated way ?

l. 100 "equval"

l. 152 "but how is it related to the actual series delta?" is not clear

l. 197 "We’ll start" informal

**Limitations:**

-

---

> ### Author Rebuttal · Authors · 2024-08-07
>
> * **“The claim that "we connect the challenge of formula creation to modern approaches in AI for Science" seems a bit bold. To my understanding this work does not involve modern AI in the sense that it requires manual and careful feature extraction and that nothing is learnt.”**
>
> In the revised manuscript, we softened this sentence in l. 234 to express a more specific claim.
>
> For context, there is no previous work that succeeded applying modern AI to conjecture generation in number theory. Our work is the first in this field to succeed with clustering algorithms, and this resulted in finding novel formulas on a large scale.
>
> *For further discussion regarding the connection of our work to modern AI for Science, please see the joint rebuttal section.*
>
> Regarding the feature extraction itself, the full set of metrics was chosen manually, because the mathematics of rational approximant series limits the possibilities of additional metrics which give independent information about the formula.
> During the conjecture generation process (Fig. 1), the subset of metrics used was selected automatically.
>
> In the revised manuscript, we now discuss the broader set of possible metrics in section 3.5, describing the relative value of including additional metrics and the rationale for the metric selection.
>
> Regarding the use of the term learning, please see Fig.5 in the attached PDF. There we show that the clusters trained on our original dataset successfully classify (based on the same dynamical metrics) new formulas outside the original dataset. The new formulas are based on higher-order polynomials than the ones used for building the clusters - showing the trained classifier can work even on higher complexity formula structures.
>
> * **“Part 3.4 was a bit confusing. The Blind-δ Algorithm is presented as a proxy for delta eq. 4. Is it not rather to estimate the approximation error epsilon ?”**
>
> This is a good point. “Blind-$\delta$” is the name we gave the entire algorithm, although this name arises from eq. 4, where indeed it is the approximation error $\epsilon$ that is being calculated without prior knowledge about the limit. It is just a terminology issue, but we should have explained it better. The ability to estimate $\delta$ without prior knowledge provides us with a unique and powerful dynamical metric, which is why we consider it a key to the algorithm.
>
> Following this comment, we clarify the use of terminology. We thank the referee for raising this point and helping to improve our work.
>
> * **“There is an inconsistency : part 3.2 it is n^beta vs table 1 P(n) and no beta ; is it because beta is hard to estimate ? is it useful for clustering ?”**
>
> We thank the referee for raising this point.
>
> The vast majority of our dataset consists of PCFs with an exponential or factorial error decay rate - in which cases the polynomial coefficient is noisy and less useful for characterizing the PCF. So yes, $\beta$ is hard to measure accurately for most PCFs.
> In fact, only 534 PCFs were measured to have both $\left| \gamma \right| < 0.1$ and $\left| \eta \right| < 0.1$ (as defined in Table 1). In those cases we believe the polynomial coefficient to be potentially useful, but a larger subset is required to test the automated process.
>
> The definitions in section 3.3 now follow the same form as Table 1, not mentioning $\beta$ in the manuscript any more.
>
> * **“Could the authors comment on the fact that the predicted delta eq. 5 can be quite far from the actual delta (fig. 2a, fig. 2b) ? Yet it seems it provides relevant information for clustering ; what does it capture ?”**
>
> We thank the referee for this remark.
>
> Figures 2a and 2b intentionally show the possible discrepancy between the direct calculation of $\delta_n$ and the value of $\delta_{\mathrm{predicted}}$ (eq. 5) for $n < 1000$. The prediction formula is accurate in the limit as $n \rightarrow \infty$. In practice, we set the numerical limit at $n = 10^7$, making $\delta_{\mathrm{predicted}}$ much closer to the actual $\delta$ value. This is now explicitly stated in the caption of fig. 2.
>
> $\delta_{\mathrm{predicted}}$ provides significant validation for the numerical $\delta$ in clustering. We used several sanity checks to identify anomalies or execution issues. One of the flags was a large discrepancy between the numerical and the predicted $\delta$.
>
> In the future, we plan to use the prediction formula in a gradient-descent-based approach to search for high-$\delta$ formulas, which we will explore in further research.
>
> The $\delta$ prediction formula holds additional importance in our research. The extension of the basic formula in Elimelech et al. (2023) to the general case was motivated and conjectured based on the results of the numerical $\delta$ measurements. In a sense, it is the first conjecture that arose from the dataset and was later proven analytically.
>
> * **“More generally, did the authors try to extract more features, other metrics, not necessarily having a mathematical interpretation, in an automated way ?”**
>
> The dynamical features are extracted automatically for the 1,543,926 formulas, but as the referee suggests, the choice of which metrics is predetermined.
>
> We have tried several other dynamic metrics during this research (like measuring $p_n$ and $q_n$ modulo primes, tracking their sign etc.) - which gave no apparent value.
>
> The idea to automate the choice of dynamic features is intriguing. Such an idea was never attempted in number theory. We thank the referee for this suggestion. It is now mentioned in Section 5 (“Discussion and Outlook”), and we leave it for future research.
>
> * **“l. 100 "equval"**
> * **l. 152 "but how is it related to the actual series delta?" is not clear**
> * **l. 197 "We’ll start" informal”**
>
> We thank the referee for the careful review and for spotting these issues.
>
> L. 100 is now corrected.
>
> L.152 is now removed.
>
> In l. 197 the first sentence is now removed.

---

> > ### Comment · Reviewer_Dgb8 · 2024-08-09
> >
> > I thank the reviewers for their replies, the precisions and the additional results they give.

---

### Official Review · Reviewer_uW6k · 2024-07-12

**Soundness:** 3
**Presentation:** 3
**Contribution:** 3
**Rating:** 7
**Confidence:** 3

**Summary:**

The paper presents a classification of 1.5 million polynomial continued fractions (PCF), continued fractions having as coefficients the integer values of two polynomials, $A(n)$ and $B(n)$, with $A$ $B$ of degree two, with integer coefficients in $[-5,5]$. PCF are classified according to the asymptotic properties of the sequence of approximation errors (difference between the convergents and the limit), the asymptotic properties of the denominator of the convergents (in simplest terms), and the measure of irrationality, which is the difference between the logs of the two previous metrics.

Along these metrics, the authors observe that groups of PCF with the same limit tend to cluster together, and that by "anchoring" these clusters on known formulas, one can discover new PCF decompositions of mathematical constants.

**Strengths:**

The paper is well written, and interesting to read. It is an original work, and both theoretical and experimental results are adequately supported. The results are intriguing, and seem to point to scientific discovery.

**Weaknesses:**

The link with AI or machine learning is not completely clear to me. Whereas the clustering techniques used in the paper, and the t-SNE projections used to represent the results, are commonly used by ML practitionners, most of the analyses conducted in this paper amount to descriptive statistical analysis of a specific mathematical dataset. To demonstrate a possible link with Machine Learning, it would be useful that the authors discuss (and perhaps demonstrate) how their approach scales to large sets of PCF, by letting the coefficients and degrees of $A$ and $B$ grow larger.

I lean towards acceptance because of the potential interest of such approaches in AI for Science.

**Questions:**

* Figure 1 caption is very long, in the interest of clarity, it might be worth describing your methodology in a specific section (3.2?)
* l.96: couldn't we assume that the convergents $p_n/q_n$ are always in simplest terms? this would avoid having to introduce of $\tilde q_n$. Besides, I believe it is assumed in the usual definition of the measure of irrationality.
* in section 3.1, you explain that the irrationality measure is either 0 or larger than 1. You then claim that the blind-$\delta$ method provides a good estimator of $\delta$, and figure 2.a seems to support this, yet in figure 2.b most estimates of $\delta$ are below $1$, and some are even negative. What happens? Doesn't this compromise the use of blind-$\delta$ as an estimator of the irrationality measure?
* Section 3.5, can you elaborate on "representation power", what it is? why is it important?
* Table 1: The exponential factor of the growth coefficient seems useless, why is it? Also, could the larger value of the Davies Boulding Index the irrationality measure be a sign of the problem with its estimation?
* Figure 4: can you provide a description of the axes in the tSNE graph? (maybe switch to PCA for more explainability)
* l.29: shouldn't Lambert's original paper be quoted, instead of a modern compilation?
* l. 100 "equal" (typo)
* l.135: the error rate $\epsilon$ is used before it is defined (in section 3.4)
* Figure 4, constant $C1$ is not defined anywhere, lemniscate?

**Limitations:**

yes

---

> ### Author Rebuttal · Authors · 2024-08-07
>
> * **“To demonstrate a possible link with Machine Learning, it would be useful that the authors discuss (and perhaps demonstrate) how their approach scales to large sets of PCF, by letting the coefficients and degrees of A and B grow larger.”**
>
> We thank the referee for this suggestion.
>
> We successfully tested our hypothesis regarding the value of identified clusters for broader forms of formulas. Specifically, we showed how the clustering applies to 3rd and 4th degree polynomial continued fractions (PCFs), showing that they belong to the same clusters created by 2nd degree PCFs, and importantly, using the same dynamical metrics. The 2nd degree PCFs in our initial dataset served as training data for a classifier (based on the automatically identified clusters) that was then tested on higher degree PCFs.
> These results are now part of a new appendix in the paper (please see the figure in the joint rebuttal PDF).
>
> * **“Figure 1 caption is very long, in the interest of clarity, it might be worth describing your methodology in a specific section (3.2?)”**
>
> We thank the referee for this suggestion.
>
> The content of Figure 1 caption is now split between the caption and the main text.
>
> * **“l.96: couldn't we assume that the convergents 𝑝𝑛/𝑞𝑛 are always in simplest terms? this would avoid having to introduce of  𝑞~𝑛.”**
>
> We thank the referee for this comment.
>
> $q̃_n$ is no longer introduced.
>
> * **“In section 3.1, you explain that the irrationality measure is either 0 or larger than 1. You then claim that the blind-𝛿 method provides a good estimator of 𝛿, and figure 2.a seems to support this, yet in figure 2.b most estimates of 𝛿 are below 1, and some are even negative. What happens? Doesn't this compromise the use of blind-𝛿 as an estimator of the irrationality measure?”**
>
> This is a delicate point in the concept of irrationality measure. Given a converging series of rational approximations $p_n / q_n$, $\delta$ is defined as the irrationality measure of that _series_ - and it can be any number $\geq$-1 (Eq.4). The irrationality measure of a _number_ is defined as the supremum of all possible $\delta$’s (Eq.3).
>
> Any converging PCF produces a $\delta$ measure - which is almost never exactly 0 or 1. But the _true_ irrationality measure of their limit will always be 0 or $\geq$1. The challenge in proving the irrationality of a mathematical constant is mostly finding a series that produces a positive $\delta$, as they are not known in advance and are notoriously hard to construct.
>
> We thank the referee for bringing this to our attention. We now stress this point in section 3.1 and reverse the order in which these two concepts are introduced.
>
> * **“Section 3.5, can you elaborate on "representation power", what it is? why is it important?”**
>
> “Representation power” of a metric can be thought of as maximal conditional information (conditioned on the metrics already included). We don’t measure information - we aim for best clustering and identification - so the added value is measured by the Davies-Bouldin Index.
>
> The core idea of step (e.2) in figure 1 is to gradually choose the metrics that give the most value for the resulting clustering of unidentified PCFs - choosing the one with the most representation power each time we need additional granularity / quality of clustering.
>
> * **“Table 1: The exponential factor of the growth coefficient seems useless, why is it?“**
>
> We believe the reason is that for the majority of PCFs the dominant convergence factor is factorial. Only when the error factorial coefficient ($\eta$) is $\approx0$, then the exponential coefficient has true meaning.
>
> The dataset contains only 72,610 PCFs with $|\eta|<0.1$. When measuring the DB Index for this subset we get:
>
> Exponential factor of the growth coefficient = 0.479275
>
> Factorial factor of the growth coefficient = 2.87425
>
> The factorial rate is now worse, as expected, but the exponential growth rate metric becomes valuable for clustering, supporting the hypothesis.
>
> * **“Also, could the larger value of the Davies Boulding Index the irrationality measure be a sign of the problem with its estimation?”**
>
> Yes, a situation where the underlying structure has good clusters but the measurement of some of the metrics is noisy can indeed produce high Davies Bouldin Index values.
>
> We believe that is not the case here:
> 1. The "Blind-$\delta$" algorithm was validated on multiple examples of PCFs with known, analytically proven, $\delta$'s.
> 2. The $\delta$ DB Index remains consistent between different random samplings of the dataset (as described in section 3.5). If the $\delta$ estimation noise was substantial enough to mask the underlying clustering, it would also create substantial variance in the clustering quality assessment.
>
>
> * **“Figure 4: can you provide a description of the axes in the tSNE graph? (maybe switch to PCA for more explainability)”**
>
> In this case there is a tradeoff between visualization and explainability. We opted to go for a better visualization.
>
> Due to the nature of tSNE, we cannot provide a simple description of the axes for the existing clustering graph, but following the referee’s comment we are now creating a new visualization based on non linear axis scales - to better combine explainability and graphical fidelity.
>
> * **“l.29: shouldn't Lambert's original paper be quoted, instead of a modern compilation?”**
>
> We thank the referee for the remark. Lambert’s work is now cited in addition to Berggren’s.
>
> * **“l. 100 "equal" (typo)**
> * **l.135: the error rate 𝜖 is used before it is defined (in section 3.4)**
> * **Figure 4, constant 𝐶1 is not defined anywhere, lemniscate?”**
>
> We thank the referee for the careful review and for spotting these issues.
>
> L. 100 is now corrected.
>
> $\epsilon$ is now defined in section 3.3.
>
> $C1$ in this context is the Continued Fraction Constant. It is now renamed to $C_{\mathrm{cf}}$ and referenced in the caption.

---

> > ### Comment · Reviewer_uW6k · 2024-08-08
> >
> > Thank you very much for your replies, which clarify a number of my questions. I will keep my rating, and believe this paper is a good fit for NeurIPS.

---

### Official Review · Reviewer_YVDn · 2024-07-12

**Soundness:** 2
**Presentation:** 3
**Contribution:** 2
**Rating:** 4
**Confidence:** 3

**Summary:**

They generate continued fraction formulas and test if they evaluate to mathematical constants.
They introduce a distance metric to compare formulas.
They discover novel formulas for known constants.

**Strengths:**

Mathematical constants are always used, so it is important to have formulas to calculate them

well-written

throughout experiments

**Weaknesses:**

It generates formula hypotheses, so we do not always know if the formulas are actually correct

Rather limited structure of the formulas

**Questions:**

all formulas are continued fraction formula of quadratic polynomials? is that not rather limited?

in (1) you give the example tan(x), but in the definition of a/b (p4 118), there is only n and no x.  So you cannot actually get a formula for tan(x), can you?

is there a list of all 1,543,926 formulas? how many are equal to pi or e? is there pi^pi or e^e among them? or ln(pi)?
do you know how many are irrational? or even how many are transcendent?

how many are proven to be correct and how many remain hypotheses?

have you consulted mathematicians if they are going to use these formulas for anything?

>(4)

is it necessary to introduce q̃_n and not use q_n without tilde there?

**Limitations:**

yes

---

> ### Author Rebuttal · Authors · 2024-08-07
>
> * **“It generates formula hypotheses, so we do not always know if the formulas are actually correct”**
> * **“how many are proven to be correct and how many remain hypotheses?”**
>
> We thank the referee for the constructive feedback.
> *Please see the joint rebuttal.*
>
> * **“Rather limited structure of the formulas”**
> * **“All formulas are continued fraction formulas of quadratic polynomials? is that not rather limited?”**
>
> First, please note the surprising generality of polynomial continued fractions (PCFs). Many useful sums, Taylor series of ubiquitous functions, and common families of integrals, are all equivalent to continued fractions via Euler’s continued fraction formula. This way, studying PCFs covers relations to trigonometric, hyperbolic, Bessel and other important functions.
> This fact is now stressed further in the revised manuscript.
>
> For more complex formulas, our dynamical metrics clustering approach can be directly extended, as it does not depend on the specific structure of the formula. Our work directly applies to a wide range of parametric families of functions: including ones whose evaluation is iterative / recursive / an infinite sum / any process producing rational approximants. Any mathematical structure of these types can be measured, clustered, and identified using the proposed method - the underlying generating functions can be thought of as a black box.
>
> To exemplify this universal concept, we also specifically looked into higher depth recursion relations, which are a promising research direction because little is known about their global structure, yet they are involved in several important conjectures. For example, the best rational approximation formula known for Euler’s gamma constant is constructed via such a recursion relation [Aptekarov et al., Trans. Moscow Math. Soc. 70, 237 (2009)]. This family of formulas is broader than continued fractions, yet our analysis showed that it is **described by the same metrics** we originally discovered for PCFs.
>
> Another type of mathematical formula that we analyzed in the revised manuscript are hypergeometric functions, which are given as infinite sums and can be analyzed by the exact same metrics that we originally developed for PCFs. Hypergeometric functions show the applicability of our approach to an even bigger family of functions for measurement, clustering, and conjecture generation. This can be useful in a wide variety of contexts, e.g., in investigations of integral formulas (e.g., Beukers-type integrals [Beukers, B. Lond. Math. Soc. 11, 268 (1979); Dougherty-Bliss et al., The Ramanujan J. 58, 973 (2022); Brown et al., arXiv:2210.03391 (2022)]).
>
> These prospects and additional potential generalizations are now expanded on in Section 5, with the above references included therein.
>
> * **“in (1) you give the example tan(x), but in the definition of a/b (p4 118), there is only n and no x. So you cannot actually get a formula for tan(x), can you?”**
>
> Yes, we can get formulas for functions of $x$ like $\tan(x)$. For this to be possible, we need $A_n,B_n$ to be functions of $n$ and of $x$.
>
> In eq. (1) the polynomials are:
> $A_n = 2n - 1$
> $B_n = -x^2$
> and the PCF converges to $x*\tan(x) - 1$.
>
> In general, given any rational $x$, there are polynomials $A_n,B_n$ with integer coefficients that produce a PCF that converges to $\tan(x)$. But the parametric family applies to any value $x$. In fig.2a, we show an example of a PCF that we found and converges to $1/\tan(1)$.
>
> Generating an $x$-dependent conjecture thus requires an additional step, but it is just as automatable as the rest. For example, once a large-enough dataset is measured and automatically identified, the family of formulas converging to $\tan(1),\tan(2)$ etc. can be extrapolated and we get a formula for $\tan(x)$.
>
> * **“Is there a list of all 1,543,926 formulas? how many are equal to pi or e? is there pi^pi or e^e among them? or ln(pi)? do you know how many are irrational? or even how many are transcendent?”**
>
> Due to its size the full dataset cannot be explicitly shown inside the article, but the code used to generate and measure it is attached to the original paper submission.
> Specifically:
>
> $\pi =$ 39 previously known + 116 new conjectures
>
> $e =$ 44 previously known + 80 new conjectures
>
> $e^2 =$ 28 previously known + 178 new conjectures
>
> Positive $\delta$ (proving irrationality) = 913,056
>
> These numbers are now stated explicitly in the manuscript.
> We thank the referee for this constructive comment.
>
> There are no $\ln(\pi)$ formulas in the dataset.
> There were no $\pi^2$ formulas in the original dataset, but we have now discovered and proved 2 new high degree PCFs for $\pi^2$ (see the PDF).
>
> Regarding the question of transcendence, there were many conjectures found for known transcendental constants (like $\pi$ and $e$) and known non-transcendental constants (like the Golden Ratio). We do not know whether the unidentified PCFs are transcendental or not.
>
> * **“have you consulted mathematicians if they are going to use these formulas for anything?”**
>
> Yes, we are in contact with experts in several fields, like Doron Zeilberger (Rutgers), Jeffrey Lagarias (University of Michigan), Uri Bader (Weizmann), Dzmitry Badziahin (University of Sydney) and others.
> We also have researchers with PhDs in mathematics as part of the team (and co-authors).
>
> One common usage is for irrationality proofs, which require a series of rational approximations with $\delta>0$. Out of the 1,543,926 converging formulas, 913,056 have $\delta>0$ and are thus irrationality proving formulas. In the next stage, we will explore how many of these constants were not known to be irrational. Such discoveries can be of substantial importance as new irrationality proofs are scarce and far between.
>
> * **“(4) is it necessary to introduce q̃_n and not use q_n without tilde there?”**
>
> We thank the referee for the suggestion.  $q̃_n$ is no longer introduced.

---

### Official Review · Reviewer_zxY9 · 2024-07-14

**Soundness:** 2
**Presentation:** 2
**Contribution:** 3
**Rating:** 5
**Confidence:** 3

**Summary:**

The paper addresses a long-standing challenge in number theory by proposing a new methodology for the categorization, characterization, and pattern identification of mathematical formulas, specifically Polynomial Continued Fraction (PCF) formulas. The authors introduce metrics based on the convergence dynamics of these formulas, enabling the first automated clustering of mathematical formulas. The methodology is demonstrated on a dataset of 1.7M PCF formulas, leading to the identification of both known and previously unknown formulas for significant mathematical constants such as π, ln(2), Gauss, and Lemniscate constants. The uncovered patterns allow for the generalization of individual formulas to infinite families, revealing rich mathematical structures. This work sets the stage for a generative model capable of creating continued fractions with specified mathematical properties, potentially accelerating the discovery of useful formulas.

**Strengths:**

The work introduces a novel methodology for the automated investigation of mathematical formulas, specifically focusing on Polynomial Continued Fractions. This approach is new in its use of convergence dynamics as metrics for clustering and categorization.

The methodology is rigorously tested on a large dataset, resulting in the discovery of both known and previously unknown formulas for important mathematical constants. This demonstrates the robustness and potential of the approach.

The paper clearly outlines the problem, the new methodology proposed, and the important findings. The inclusion of detailed explanations and relevant figures helps in understanding the approach and its implications.

By automating the discovery of mathematical formulas, this work has the potential to impact the field of number theory and mathematical discovery. The ability to generalize formulas into infinite families could lead to new insights and advancements in mathematics.

**Weaknesses:**

The paper has some weaknesses that should be addressed to improve its quality:


While the addressed task and the methodology is unique, it may not be the best fit for a machine learning conference like NeurIPS. The focus on mathematical discovery might be better suited for a specialized conference or journal in mathematics or computational mathematics.

The paper’s writing can be further improved for clarity and conciseness. The abstract could be more concise, and the figures are currently blurry and not well-organized, detracting from the overall presentation.

The study is based on a limited-size dataset and a small set of metrics. Expanding the dataset and incorporating a broader range of metrics could enhance the robustness and applicability of the findings.

The newly identified formulas for significant constants need to be further verified. Ensuring their correctness and utility is crucial for the validity of the contributions.

**Questions:**

How do the authors plan to further verify the newly discovered formulas for significant constants such as π and ln(2)?

Can the authors provide more details on how their methodology can be scaled to larger datasets and more complex formulas?

How can the approach be adapted or extended to other types of mathematical formulas beyond Polynomial Continued Fractions?

**Limitations:**

The paper could benefit from a more detailed discussion on the impact of the limited dataset size and the potential benefits of using larger and more diverse datasets.

The choice of metrics is crucial for the clustering and characterization of formulas. Providing a rationale for the selected metrics and discussing potential additional metrics would strengthen the paper.

Further elaboration on the methods for verifying the newly discovered formulas would enhance the credibility of the findings.

---

> ### Author Rebuttal · Authors · 2024-08-07
>
> * **“While the addressed task and the methodology is unique, it may not be the best fit for a machine learning conference like NeurIPS. The focus on mathematical discovery might be better suited for a specialized conference or journal in mathematics”**
>
> *Please see the joint rebuttal about this important point.*
>
> * **“The paper’s writing can be further improved for clarity and conciseness. The abstract could be more concise, and the figures are currently blurry and not well-organized, detracting from the overall presentation.”**
>
> We thank the referee for this constructive feedback.
> The abstract has been shortened and made more concise, and the figures' quality and readability improved.
>
> * **“The _choice of metrics_ is crucial for the clustering and characterization of formulas. Providing a _rationale for the selected metrics_ and discussing _potential additional metrics_ would strengthen the paper.”**
> * **“...and a _small set of metrics_. Expanding the dataset and _incorporating a broader range of metrics_ could enhance the robustness and applicability of the findings.”**
>
> Indeed, the choice of metrics is crucial. We have tried several other metrics during this research (like measuring $p_n$ and $q_n$ modulo primes), which gave no apparent value.
> In practice, our selection is automated (Fig. 1), selecting the metrics that provide the largest immediate improvement in clustering.
>
> In the revised manuscript, we now discuss the broader set of possible metrics in Section 3.5, describing the relative value of including additional metrics and the rationale for the metric selection.
>
> We thank the referee for this constructive and useful comment.
>
> * **“The paper could benefit from a more detailed discussion on the impact of the _limited dataset size_ and the potential benefits of _using larger_ and more diverse _datasets_.”**
> * **“Can the authors provide more details on how their methodology can be _scaled to larger datasets_…?”**
> * **“The study is based on a _limited-size dataset_…“**
>
> Our methodology can be scaled to larger datasets in multiple ways. For example, extend the range of PCF coefficients from [-5,5] to [-10,10], increasing the size of the dataset ~50X, or going up to 3rd degree $A_n$ and 4th degree $B_n$ with coefficients in [-5,5], increasing the dataset ~1333X.
>
> To combat this rapid growth in required compute, we propose a 2-fold approach:
> * Make the measurement depth dynamically chosen (instead of constant for all) during the evaluation - aiming at a fixed precision for all PCFs instead of a fixed fraction depth.
> * The heaviest part of the computation - calculating the metrics for each formula - is “embarrassingly parallelizable”. We recently adapted our algorithm to the Berkeley Open Infrastructure for Network Computing (BOINC), enabling parallel experiments on thousands of volunteer computers. Assuming a typical contribution of 1000 BOINC volunteer cores, we expect the above dataset to require about ~1 month of compute.
>
> These estimations and advances are now added to Appendix A.
>
> * **“Can the authors provide more details on how their methodology can be scaled to … _more complex formulas_?”**
> * **“How can the approach be adapted or _extended to other types of mathematical formulas_ beyond Polynomial Continued Fractions?”**
>
> First, please note the surprising generality of PCFs. Many useful sums, Taylor series and families of integrals, are all equivalent to continued fractions via Euler’s formula (see PDF). This way, studying PCFs covers relations to trigonometric, hyperbolic, Bessel and other important functions.
>
> This fact is now stressed further in the revised manuscript.
>
> Our dynamical metrics clustering methodology can be scaled to more complex formulas as it does not depend on the specific structure of the underlying function - including ones whose evaluation is iterative / recursive / an infinite sum / any process producing rational approximants. Any such mathematical structure can be measured, clustered, and identified using the proposed method - treating the generating functions as a black box.
>
> To exemplify this universal concept, we also specifically looked into higher depth recursion relations, which are a promising research direction because little is known about their global structure, yet they are involved in several important conjectures. For example, the best rational approximation formula known for Euler’s gamma constant is constructed via such a recursion relation [Aptekarov et al., Trans. Moscow Math. Soc. 70, 237 (2009)]. This family of formulas is broader than continued fractions, yet our analysis showed that it is **described by the same metrics** we originally discovered for PCFs.
>
> Another type of mathematical formula that we analyzed in the revised manuscript are hypergeometric functions, which are given as infinite sums and can be analyzed by the exact same metrics that we originally developed for PCFs. Hypergeometric functions show the applicability of our approach to an even bigger family of functions for measurement, clustering, and conjecture generation. This can be useful in a wide variety of contexts, e.g., in investigations of integral formulas (e.g., Beukers-type integrals [Beukers, B. Lond. Math. Soc. 11, 268 (1979); Dougherty-Bliss et al., The Ramanujan J. 58, 973 (2022); Brown et al., arXiv:2210.03391 (2022)]).
>
> These prospects and additional potential generalizations are now expanded on in Section 5, with the above references included therein.
>
> * **“The newly identified formulas for significant constants need to be further verified. Ensuring their correctness and utility is crucial for the validity of the contributions.”**
> * **“How do the authors plan to further verify the newly discovered formulas for significant constants such as π and ln(2)?”**
> * **“Further elaboration on the methods for verifying the newly discovered formulas would enhance the credibility of the findings.”**
>
> *Please see the joint rebuttal about this important point.*

---

### Author Rebuttal · Authors · 2024-08-07

We would like to summarize and address the most important comments brought by more than one referee.

Regarding the link of our work to ML
---

* **“While the addressed task and the methodology is unique, it may not be the best fit for a machine learning conference like NeurIPS.”**
* **“The link with AI or machine learning is not completely clear to me.”**

Until now, leading methods in ML were not successful in problems of conjecture generation in number theory. This field has been harder to penetrate compared to other areas, such as theorem proving. Our work is the first to successfully apply a basic learning method to conjecture generation in number theory. This is also the first example of successful clustering for conjecture generation in any area of mathematics. Our clustering methods are well-known in ML, but the underlying dynamical metrics we found and used are completely new and they open a path for many other ML applications in this field.

For context, our work offers advances in automated conjecture generation (ACG), a subfield of AI for Science. Notably, current generation LLMs do not succeed in generating relevant new conjectures in number theory. The reason is attributed to the lack of metrics, needed to provide a measure of being “closer to correct”. In number theory, such metrics have not existed until now - our work is the first to suggest successful metrics, and to use them.

In the time that passed since our initial submission, we improved our manuscript especially in these aspects. We successfully tested our clustering method on broader forms of formulas and found clusters applicable to formulas in ranges of parameters outside those used for training: 3rd and 4th degree polynomial continued fractions (PCFs) are captured by the same clusters trained on (constructed by) 2nd degree PCFs, and importantly, using the same dynamical metrics. These results are now part of the new appendix D (see the revised Fig. 2b in the attached PDF).

Our systematic approach for formula generation, with these new validations, provided the first example of automated learning for conjecture generation in number theory, hopefully inspiring other efforts for ML applications in this field.

Regarding the correctness of the automatically generated formulas
---

* **“The newly identified formulas for significant constants need to be further verified. Ensuring their correctness and utility is crucial for the validity of the contributions”.**
* **“How do the authors plan to further verify the newly discovered formulas for significant constants such as π and ln(2)?”**
* **“Further elaboration on the methods for verifying the newly discovered formulas would enhance the credibility of the findings.”**
* **“It generates formula hypotheses, so we do not always know if the formulas are actually correct”**
* **“how many are proven to be correct and how many remain hypotheses?”**

Following these remarks, we have taken two important verification steps:

1. The novel formulas are now evaluated to a higher depth: 4 million steps or more, producing between 13 digits to thousand of digits of precision, depending on the convergence rate. These additional digits reduce the chance of an incorrect accidental identification by a measure independent of our clustering (elaborated in the revised Appendix A). Even with this additional verification, none of the previously identified constants was found erroneous, further supporting the robustness of our approach.

2. We worked to prove selected formulas in cases of especially slow convergence (see the attached PDF). 47 of the automatically generated conjectures are now analytically proven. These proofs help validate our approach and show that the results can be relied on in future research efforts. We are working in parallel with mathematicians on general mathematical approaches for proofs that can be applied in scale, and automatically, which is necessary to cope with the large number of newly discovered formulas.

In addition, we emphasize that the value of unproven formulas (or generally conjectures) cannot be understated, as it is usually such a conjecture that acts as the first step that eventually leads to a discovery of a new theory. For example, consider Srinivasa Ramanujan’s contributions to mathematics, many of which were initially unproven formulas, yet his impact on the mathematical world is undeniable. In a similar way, our formula generation algorithm provides new leads for mathematical research that can have long-term impact.

---

### Decision · Program_Chairs · 2024-09-25

**Decision:**

Accept (poster)

**Comment:**

This paper presents an original contribution to automatized discovery of mathematical formulas. While the machine learning methods it uses are rather basic, it does an extensive job of evaluating a large number of continued fractions. It discovers previously unknown (at least to the authors and the reviewers) formulas. This is a valuable contribution to the field and will be of interest to the NeurIPS community.

The reviews contain many points that improve the presentation of the paper. The authors are strongly encouraged to carefully take them into account in order to increase the impact of their interesting work.